# Contrastive Residual Energy Test-time Adaptation

## Abstract

Test-Time Adaptation (TTA) enhances model robustness by enabling adaptation to target distributions that differ from training distributions, improving real-world generalizability. However, most existing TTA approaches focus on adjusting the conditional distribution and therefore exhibit poor calibration, as they rely on uncertain predictions in the absence of labels. Energy-based TTA frameworks provide an alternative by modeling the marginal distribution of target data without depending on label predictions, but their reliance on costly sampling hinders scalability in real-world scenarios where decisions must be made without latency. In this work, we propose *Contrastive Residual Energy Test-time Adaptation (*CRETTA*)*, a practical solution for reliable adaptation. We first redefine the marginal distribution of target data using residual energy function and embed it into contrastive objective. This design prevents overfitting through adaptive gradient reweighting mechanism that leverages the relative residual energy while eliminating the sampling process. Extensive experiments demonstrate that CRETTA achieves scalable and well-calibrated adaptation under real-world computational constraints.

## 1 Introduction

Deep learning models can achieve high accuracy on training and testing data from the same distribution. However, when the distribution of the test data diverges from the original training dataset, the performance of the deep learning models deteriorates. This *distribution shift* refers to changes in the underlying data statistics, such as feature distributions or environmental conditions, between training and deployment. It is a major challenge in real-world scenarios, where test samples are often drawn from a distribution that deviates from the training data.

To address distribution shifts during testing, *test-time adaptation* (TTA) strategy aims to adapt trained model instantly, thereby maintaining robust performance on unexpected out-of-distribution samples. Since ground-truth labels are unavailable at test-time, existing approaches such as Pseudo-labeling (Lee et al., 2013) and SHOT (Liang et al., 2020) use the model's own predictions as pseudo-labels. Likewise, TENT (Wang et al., 2020) operates similarly by using the model's predicted probability distribution as a surrogate ground-truth distribution within an entropy-minimization objective.

Formally, the entropy minimization objective for a test sample $x_t$ is expressed as $-\sum_{k=1}^{C} p_\theta(\hat{y}_k \mid x_t) \log p_\theta(\hat{y}_k \mid x_t)$, where $C$ is the number of classes and $p_\theta(\hat{y}_k \mid x_t)$ is the predicted probability of class $\hat{y}_k$. While this approach demonstrates promising accuracy, it relies on uncertain predictions without ground-truth supervision. As a result, optimizing the entropy minimization objective often drives the predicted probabilities to collapse to extreme values of 0 or 1, leading to overconfident predictions (Press et al., 2024). This behavior increases calibration error as illustrated in Figure 10. In high-stakes real-world scenarios, such overconfidence can be detrimental, underscoring the need for alternative TTA strategies beyond simple entropy minimization. In such settings, *well-calibrated adaptation is critical to avoid overconfident errors and ensure safe, reliable model behavior in real-world scenarios.*

**Adaptation with marginal distribution leads to better calibration.** Rather than learning from the conditional distribution $p_\theta(\hat{y}|x)$ with unreliable model predictions, some approaches instead focus on modeling the marginal distribution $p_\theta(x)$ by directly maximizing its likelihood. This formulation avoids dependence on predicted labels and mitigates the overconfidence issues often associated with

entropy minimization. TEA (Yuan et al., 2024) applies maximum likelihood estimation (MLE) in the TTA setting and leverages contrastive divergence (CD) loss, an energy-based objective (Hinton, 2002; LeCun et al., 2006; Song & Kingma, 2021), to align the model with unseen target distributions. Achieving better calibration while maintaining strong classification accuracy, TEA provides experimental evidence of stable and reliable adaptation, extending beyond the theoretical insights of earlier works (He et al., 2021; Wang et al., 2021; Schröder et al., 2024).

Despite its strengths, energy-based models with MLE methods for TTA still face two critical limitations: **(i) Unresolvable approximation error:** The normalization approximation relies on short-run Markov chains, which yield biased gradient estimates. As a result, parameter updates at test time can be unstable or converge to suboptimal solutions (Song & Kingma, 2021; Yair & Michaeli, 2021), particularly during few-shot adaptation or when dealing with high-dimensional data as shown in Figure 1. **(ii) High computational cost due to sampling-based approximation:** It requires repeated sampling to re-estimate the normalization constant for every incoming test batch, incurring substantial computational overhead during adaptation. This burden grows with model sizes, since each sampling step entails expensive gradient computations, thereby making real-time or resource-constrained TTA deployment infeasible, highlighting the need for methods that reduce overhead while maintaining strong performance.

To this end, we propose CRETTA, a novel residual energy-based test-time adaptation method that optimizes with marginal distribution while eliminating the normalization constant approximation, achieving high computational efficiency and scalability for real-world deployment.

First, we redesign the TTA task as learning the residual component of the distribution shift that the pretrained model has not yet captured and model the discrepancy with a residual energy function. Then, we embed the residual energy function into the contrastive learning objective, offsetting the normalization constant for the marginal distribution which typically requires extensive computations for approximation.

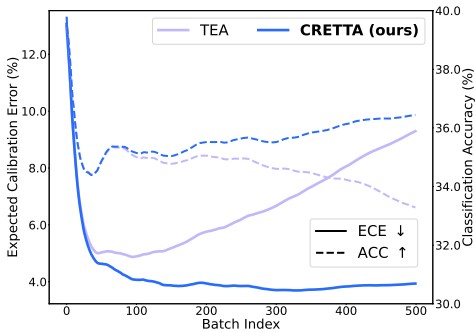

Figure 1: ImageNet-C (Sev 5) ECE($\downarrow$) and Acc($\uparrow$) over batch progress. CRETTA maintains stable calibration performance, while TEA experiences approximation error accumulation.

The primary contributions of our work can be summarized as follows:

- We introduce CRETTA, a novel sampling-free residual energy-based test-time adaptation (TTA) framework, offering computational efficiency and scalability for practical applications.

- Our method mathematically redefines the target marginal distribution with a residual energy function and optimizes it with contrastive learning objective. Consequently, this formulation enables TTA without costly normalization constant approximation and mitigates overfitting, thereby ensuring stable adaptation.

- CRETTA is well-calibrated and achieves strong performance across various distribution shifts.

## 2 PRELIMINARIES

**Problem Setup** Let $Q$ denote the marginal distribution of the source training data $x_s$. Consider a classifier $f_\phi(x)$ with parameters $\phi$, which is pretrained on a labeled source dataset $\{(x_s^{(i)}, y_s^{(i)})\}_{i=1}^M$. Although this pretrained model performs well on in-distribution test data (i.e., $x_s \sim Q$), its performance can degrade substantially when tested on data from a different distribution $P(\neq Q)$, commonly referred to as out-of-distribution data.

Test-time adaptation (TTA) aims to mitigate this issue by adapting the pretrained parameters $\phi$ to better align with the target marginal distribution $P$. In this setting, we are given a set of $N$ unlabeled target samples $\{x_t^{(i)}\}_{i=1}^N$ drawn from $P$, which arrive in online batches. To cope with the absence of label information, existing methods often rely on unsupervised objectives, particularly entropy minimization. EATA (Niu et al., 2022) and SAR (Niu et al., 2023) build on this foundation

by incorporating surrogate objectives and sample selection mechanisms to filter out unreliable predictions. However, these enhancements still fundamentally depend on uncertain model outputs, as dictated by the nature of entropy-based objectives.

Moving beyond entropy-based methods, recent work by TEA (Yuan et al., 2024) demonstrates the promise of energy-based modeling for TTA, where the central idea is to represent the test data marginal distribution through an energy function. Within this framework, TEA employs MLE on the marginal distribution of test samples $\{x_t^{(i)}\}_{i=1}^N$, so that the energy function is learned to assign lower energy (i.e., higher likelihood) to observed test inputs. By directly modeling and adapting to the marginal distribution, TEA mitigates distribution shifts without requiring labeled data or relying heavily on potentially unreliable model predictions. Building on this insight, our proposed method, CRETTA, approaches test-time adaptation not merely as an MLE procedure, but as effective learning of an energy function that represents an unnormalized marginal distribution.

**Energy-based Models (EBMs)**  LeCun et al. (2006) express the marginal distribution in the energy-based model framework using Gibbs distribution, which can be formulated as $q_\phi(x) = \exp(-E_\phi(x))/Z(\phi)$, where $\phi \in \Phi$, with $\Phi$ representing the parameter space and $Z(\phi) = \int_x \exp(-E_\phi(x))dx$ is the normalizing constant (partition function). The energy function $E_\phi(x) : \mathbb{R}^d \to \mathbb{R}$ maps a $d$-dimensional data point to a scalar energy value, thereby defining an unnormalized density over the data space. The fundamental principle of EBMs is to represent the likelihood of a sample through this energy landscape: lower energy corresponds to higher likelihood and vice versa. A well-trained EBM thus learns to assign low energy values (i.e., high likelihood) to samples drawn from the in-distribution (source) $Q$, while assigning high energy (i.e., low likelihood) to out-of-distribution samples, such as those from a shifted target distribution $P$, where $P \neq Q$.

Grathwohl et al. (2019) and Yang & Ji (2021) present an innovative perspective on reframing discriminative models within the EBM framework. In their formulation, the energy function for a given input-label pair $(x, y)$ is defined as $E_\phi(x, y) = -f_\phi(x)[y]$, where $f_\phi(x)[y]$ denotes the logit corresponding to label $y$ in the discriminative model $f_\phi$ (i.e., classifier). Furthermore, the energy function derived from a discriminative model for a single input $x$ can be expressed as the negative `log-sum-exp` of the logits across all classes in the final classifier layer:

$$E_\phi(x) = -T \cdot \log \sum_{k=1}^{C} e^{f_\phi(x)[k]/T},\tag{1}$$

where $T$ is a temperature parameter that controls the sharpness of the distribution (Liu et al., 2020). Finally, using a discriminative model within the EBM framework allows one to express the marginal probability of a data sample $x$. The main challenge in optimization stems from the normalization constant $Z$, which requires integrating over the entire input space, a task that is generally intractable. Consequently, EBMs often rely on specialized training methods such as contrastive divergence (Carreira-Perpinan & Hinton, 2005) or Markov chain Monte Carlo (MCMC) sampling to approximate or avoid the direct computation of $Z$.

## 3  METHODS

**A Residual Perspective on Distribution Shift**  To characterize the distribution shift at test-time, we utilize a residual energy function that captures the discrepancy between the source and target distributions. Formally, let $Q$ denote the source distribution and $P$ the target distribution. We can express $P$ in terms of $Q$ via an exponential factor encoding the residual energy: $P = Q \exp(-R)/Z$, where $R$ is a residual energy function, and $Z$ is a normalization constant.

By analogy, the marginal distribution of the target data $p_\theta$ can be written as the product of the pretrained source model $q_\phi$ and an exponential residual term:

$$p_\theta(x) = \frac{1}{Z(\theta)} q_\phi(x) \exp\left(-\frac{1}{\beta}\tilde{E}_\theta(x)\right),\tag{2}$$

where the residual energy function $\tilde{E}_\theta$ is designed to model only the discrepancy between the fixed source model and the target distribution. Moreover, $\beta > 0$ is a temperature parameter and $\log Z(\theta)$ is constant across samples. During TTA, the source model $q_\phi$ remains frozen, and $\tilde{E}_\theta$ is learned to

capture only the distributional differences that arise under domain shift. In other words, $\tilde{E}_\theta$ focuses exclusively on the distribution-shift-induced residuals, refining the energy landscape so that the combined model aligns more closely with the true target distribution while preserving the original source knowledge.

Our residual interpretation of distribution shift can be viewed as an extension of the architectural constraints commonly employed in standard TTA setup. Building on the observation that updating only a subset of model parameters, such as the batch-normalization (BN) layers, enables efficient and stable adaptation by mitigating overfitting to severe distribution changes (Wang et al., 2020; Wu et al., 2024; Zhao et al., 2023), numerous TTA methods adopt such restricted update strategies. From a mathematical perspective, we extend this idea by recasting TTA as the problem of learning the residual component of the distribution shift that the pretrained model has not yet captured. This formulation serves as an implicit regularizer: it constrains the target model to learn only the unmodeled portion of the shift, thereby limiting deviation from the source distribution and preventing overfitting, as discussed in subsection 4.3.

**Learning the Residual Energy via Contrastive Objective**   Energy-based models (EBMs) trained with maximum-likelihood estimation (MLE) often suffer from *biased gradients and prohibitive sampling costs*, primarily due to the need to approximate the intractable partition function $Z$ (Song & Kingma, 2021). These limitations render conventional MLE approaches fundamentally ill-suited for practical TTA, where efficiency and stability are critical.

To overcome these challenges, we propose a *contrastive learning* framework that directly learns the residual energy function without any estimation or approximation of the partition function $Z$. Instead of optimizing likelihoods, we operate entirely on pairwise energy differences between source and target samples. This eliminates the need for sampling from the model distribution, making our method both tractable and scalable for TTA.

Our method shares conceptual similarities with Noise Contrastive Estimation (NCE) (Gutmann & Hyvärinen, 2010), in that both use contrastive objectives to bypass normalization. However, unlike NCE, which retains an implicit dependence on $Z$ through the requirement of globally normalized densities, our formulation dispenses with $Z$ entirely, as it only requires relative energies for learning.

Formally, we reinterpret the residual energy function $\tilde{E}_\theta(x)$ as arising from the density ratio between the target distribution $p_\theta(x)$ and the fixed source model $q_\phi(x)$:

$$\tilde{E}_\theta(x) = -\beta \left( \log \frac{p_\theta(x)}{q_\phi(x)} + \log Z(\theta) \right).$$

Assuming that the residual energy function $\tilde{E}_\theta$ should favor target samples $x_t$ over source samples $x_s$ (i.e., assigning lower energy to $x_t$ than to $x_s$), we model the probability that the residual energy of a target sample is lower than that of a source sample, as:

$$P\left( \tilde{E}_\theta(x_t) < \tilde{E}_\theta(x_s) \right) = \frac{1}{1 + \exp\left( -\tilde{E}_\theta(x_s) + \tilde{E}_\theta(x_t) \right)} = \frac{1}{1 + \exp\left( \beta \log \frac{p_\theta(x_s)}{q_\phi(x_s)} - \beta \log \frac{p_\theta(x_t)}{q_\phi(x_t)} \right)},$$

During optimization, this objective drives the residual energy function $\tilde{E}_\theta$ to consistently reflect the distribution shift by lowering the relative energy of target samples with respect to source samples, thereby aligning the model with the target distribution while keeping the source model fixed.

Finally, we derive the optimization objective for learning the target model $p_\theta$ with the source model $q_\phi$. Given a set of source and target pairs $(x_s, x_t)$, the objective can be formulated as minimizing the negative log-likelihood of the probability:

$$\mathcal{L}(\theta; \phi, \mathcal{B}) = -\frac{1}{|\mathcal{B}|} \sum_{(x_s, x_t) \sim \mathcal{B}} \log \sigma \Big( \beta \underbrace{\left( \log \frac{p_\theta(x_t)}{q_\phi(x_t)} - \log \frac{p_\theta(x_s)}{q_\phi(x_s)} \right)}_{l} \Big),$$

$$\text{where } l = -(E_\theta(x_t) - E_\theta(x_s)) + (E_\phi(x_t) - E_\phi(x_s)).$$

(3)

Crucially, as a consequence of our pairwise contrastive objective, the partition function $Z$ cancels out completely, and no sampling is required unlike in MLE or NCE settings. Instead, we leverage a minimal buffer of source samples $\mathcal{B}_s = \{x_s^{(i)} \mid i = 1, \ldots, |\mathcal{B}_s|\}$, to guide optimization and enable stable contrastive adaptation under test-time distribution shift. Despite its critical role in stabilizing optimization, the buffer size is negligibly small, imposing virtually no burden in modern memory settings and introducing no practical limitations in real-world deployment.

To optimize our objective (Equation 3), we construct a pairwise mini-batch $\mathcal{B} = \{(x_s^{(i)}, x_t^{(i)})\}$, where each pair consists of a target sample $x_t^{(i)} \in \mathcal{B}_t$ from the current target stream and a corresponding source sample $x_s^{(i)} \in \mathcal{B}_s$ randomly drawn from the source buffer $\mathcal{B}_s$. We demonstrate that our method maintains consistent performance even when the source buffer size $|\mathcal{B}_s|$ is reduced to as little as $1\%$ of the source dataset, significantly lowering memory overhead. The robustness of our approach is further validated through an ablation study, as presented in Table 5.

Our proposed objective effectively aligns the model with the target distribution while avoiding the explicit estimation of the residual energy function. Furthermore, we reformulate both the source and target models as energy-based models, denoted as $E_\phi$ and $E_\theta$, respectively, following Equation 1, with the target one initialized as $\theta = \phi$. This reformulation allows us to express the objective solely in terms of energy functions, eliminating the need for explicit normalization constants through algebraic simplifications. For a detailed derivation, we provide the full mathematical formulation in subsubsection A.1.1.

**Why Does Contrastive Residual Learning Yield Stable Adaptation?** The stable adaptation achieved by contrastive residual learning can be clarified through a gradient analysis. The gradient of our objective in Equation 3 is computed as follows:

$$
\nabla_\theta \mathcal{L}(\theta; \phi, \mathcal{B}) = -\frac{1}{|\mathcal{B}|} \sum_{(x_s, x_t) \sim \mathcal{B}} \beta \cdot w(x_t, x_s) \cdot \left( \nabla_\theta \log p_\theta(x_t) - \nabla_\theta \log p_\theta(x_s) \right),
$$

$$
\text{where } w(x_t, x_s) = \sigma \left( \beta \log \frac{p_\theta(x_s)}{q_\phi(x_s)} - \beta \log \frac{p_\theta(x_t)}{q_\phi(x_t)} \right)
$$

$$
= \sigma \left( \beta \cdot \left( E_\theta(x_t) - E_\phi(x_t) \right) - \beta \cdot \left( E_\theta(x_s) - E_\phi(x_s) \right) \right). \tag{4}
$$

In this context, the term *contrastive* does not merely imply decreasing the energies of target samples or increasing those of source samples. Rather, the gradient weights are modulated by the relative energy levels of paired source-target samples, which promotes stable adaptation (see subsection 4.3 for further analysis).

If we remove the residual assumption, the pairwise contrastive learning objective reduces to the form in subsection C.1. In this case, there are no bias terms parameterized by $\phi$, so the gradient magnitude depends solely on the target model's energies, making the method more prone to overfitting, as illustrated in Figure 3. A more detailed mathematical discussion of the residual assumption and the pairwise contrastive approach is provided in Appendix B and Appendix C.

## 4 EXPERIMENT

In this section, we present a comprehensive analysis of our proposed method, CRETTA, and conduct a detailed comparison against state-of-the-art approaches using widely adopted benchmark datasets.

### 4.1 EXPERIMENTAL SETUP

**Benchmark Datasets and Metrics** To evaluate corruption robustness in test-time adaptation, we selected three benchmark datasets: (i) **CIFAR10-C**, (ii) **CIFAR100-C**, and (iii) **TinyImageNet-C** (Hendrycks & Dietterich, 2019). Each dataset contains 15 unique corruption types, categorized into 5 severity levels. In our evaluation, we reported the performance as the average across all 15 corruption types to provide a comprehensive measure of robustness. To rigorously evaluate the practical applicability of the proposed TTA method, we used three evaluation metrics: (i) **Accuracy (ACC)**, (ii) **Expected Calibration Error (ECE)**, and (iii) **Giga Floating-Point Operations (GFLOPs)**.

Table 1: Comparison of classification accuracy (Acc ↑) and expected calibration error (ECE ↓) on the CIFAR10-C, CIFAR100-C, and TinyImageNet-C datasets at corruption severity level 5 and the average across severity levels 1-5. The best results are emphasized in **BOLD**, while the second-best results are UNDERLINED.

| | CIFAR-10-C | | | | CIFAR-100-C | | | | TinyImageNet-C | | | |
| | Severity L5 | | Severity Avg | | Severity L5 | | Severity Avg | | Severity L5 | | Severity Avg | |
| Method | Acc(↑) | ECE(↓) | Acc(↑) | ECE(↓) | Acc(↑) | ECE(↓) | Acc(↑) | ECE(↓) | Acc(↑) | ECE(↓) | Acc(↑) | ECE(↓) |
|---|---|---|---|---|---|---|---|---|---|---|---|---|
| Source | 81.73% | 10.18% | 88.82% | 5.45% | 53.25% | 17.71% | 64.11% | 11.73% | 35.12% | 16.17% | 43.16% | 13.46% |
| *Normalization* | | | | | | | | | | | | |
| BN Adapt | 85.46% | 4.85% | 89.12% | 3.15% | 60.74% | 8.32% | 65.83% | 6.88% | 39.60% | 13.66% | 44.72% | 12.12% |
| *Pseudo Labeling* | | | | | | | | | | | | |
| PL | 84.85% | 10.10% | 90.09% | 6.20% | 56.33% | 23.81% | 65.72% | 16.66% | 35.40% | 30.95% | 43.79% | 23.47% |
| SHOT | 87.91% | 5.42% | 90.78% | 3.86% | 64.41% | 8.93% | 68.80% | 7.44% | 39.84% | 13.81% | 44.95% | 12.24% |
| *Entropy Minimization* | | | | | | | | | | | | |
| TENT | 87.84% | 5.49% | 90.74% | 3.89% | 64.31% | 8.93% | 68.73% | 7.47% | 39.83% | 13.82% | 44.94% | 12.24% |
| ETA | 85.46% | 4.85% | 89.12% | 3.15% | 61.77% | 8.54% | 66.66% | 7.10% | 39.67% | 13.70% | 44.79% | 12.16% |
| EATA | 85.46% | 4.85% | 89.12% | 3.15% | 61.79% | 8.54% | 66.65% | 7.11% | 39.68% | 13.70% | 44.79% | 12.16% |
| SAR | 86.54% | 4.79% | 89.80% | 3.13% | 62.71% | 8.31% | 67.36% | 6.91% | 39.66% | 13.72% | 44.77% | 12.16% |
| AEA | 88.27% | 5.09% | 90.88% | 3.73% | 64.40% | 9.16% | 68.75% | 7.61% | 39.87% | 13.82% | 44.97% | 12.25% |
| *Energy-based Models* | | | | | | | | | | | | |
| TEA | 88.06% | **3.83%** | 90.67% | **2.68%** | 63.66% | **7.68%** | 67.93% | **6.33%** | 39.96% | 13.84% | 45.08% | 12.24% |
| CRETTA | **88.30%** | 4.15% | **91.01%** | 2.88% | **64.52%** | 7.99% | **69.05%** | 6.82% | **40.30%** | **13.52%** | **45.75%** | **11.85%** |

**Baselines** We compared our method with state-of-the-art approaches. (i) **Source:** The pre-trained classifier from the source data which performs inference on test data without adaptation. (ii) **Normalization-based:** BN-Adapt (Schneider et al., 2020) updates batch normalization statistics for test samples. (iii) **Pseudo-labeling-based:** Pseudo-Labeling (PL) (Lee et al., 2013) and SHOT (Liang et al., 2020) where test samples are filtered based on a confidence threshold, and the model is optimized using these pseudo-labels. (iv) **Entropy-based:** TENT (Wang et al., 2020), ETA, EATA (Niu et al., 2022), SAR (Niu et al., 2023), and AEA (Choi et al., 2025) aim to minimize entropy on test samples to achieve alignment with the target distribution. (v) **Energy-based:** TEA (Yuan et al., 2024) adapts to the marginal probability of the target distribution using energy-based learning with SGLD sampling.

**Implementation Details** In our experiments, we employed WRN-40-2 (Zagoruyko, 2016) for CIFAR10-C and CIFAR100-C datasets, and WRN-28-10 for TinyImageNet-C as backbones. Pre-trained weights were sourced from RobustBench (Croce et al., 2020). If unavailable, models were trained from scratch. We conduct online adaptation and evaluation following TENT (Wang et al., 2020) and TEA (Yuan et al., 2024) employing the Adam optimizer (Kingma, 2014) and reported results over three different random seeds. To further enhance robustness during adaptation, we incorporated data augmentation into the source buffer, which contained only 10% of the original source dataset. For more detailed information, please refer to the appendices.

## 4.2 PERFORMANCE COMPARISON

**Classification Accuracy and Calibration Error** Table 1 reports accuracy, focusing on the highest severity level 5 and the average across severity levels (1-5) across all datasets and corruption severities. Our proposed method consistently outperformed all other baselines under corrupted settings, notably achieving accuracy of 40.30% at the highest severity (level 5) on TinyImageNet-C. This consistent improvement highlights CRETTA's adaptability and effectiveness in handling larger and more complex datasets, reinforcing its suitability for real-world test-time adaptation scenarios.

In TTA, model calibration is crucial for quantifying the prediction uncertainty, ensuring reliability under domain shifts and unlabeled data scenarios. We evaluated calibration performance using Expected Calibration Error (ECE) with 10 bins. While TEA performs well on CIFAR datasets, it fails to maintain the same level of superiority on TinyImageNet-C. In contrast, CRETTA demonstrates strong overall performance across all datasets (Table 1). Specifically, on TinyImageNet-C, CRETTA consistently outperforms other methods on most of corruption types in calibration as reported in Table 9.

**Scalability** We further evaluate CRETTA on PACS and ImageNet-C datasets. On PACS, CRETTA maintains competitive classification accuracy (Table 2) while achieving the lowest average ECE, outperforming the entropy-based method TENT and the existing energy-based method TEA by a significant margin (Table 3). On ImageNet-C, CRETTA achieved substantially lower ECE(2.69%) with higher accuracy, whereas TEA's ECE was 7.21% (Table 12).

We interpret TEA's weaker performance on both datasets as a consequence of approximation errors when estimating the normalization constant during sampling. These results underscore that CRETTA generalizes well to large-scale, style shifted datasets, achieving strong predictive performance and superior calibration.

Table 2: Classification accuracy (Acc ↑) on PACS.

| Source Domain | Method | Target Domain | | | | Avg |
|---|---|---|---|---|---|---|
| | | Photo | Art | Cartoon | Sketch | |
| Photo | Source | - | 55.39 | 22.61 | 23.17 | 33.72 |
| | TENT | - | 57.89 | 63.82 | 32.84 | 51.52 |
| | TEA | - | 53.16 | 50.33 | **33.17** | 45.55 |
| | CRETTA | - | **64.45** | **64.83** | 29.89 | **53.06** |
| Art | Source | 88.58 | - | 49.26 | 35.06 | 57.64 |
| | TENT | **92.10** | - | 63.18 | 35.23 | 63.50 |
| | TEA | 83.95 | - | 53.38 | 36.52 | 57.95 |
| | CRETTA | 91.52 | - | **65.90** | **40.99** | **66.13** |
| Cartoon | Source | 70.52 | 62.52 | - | 50.44 | 61.16 |
| | TENT | 82.34 | 64.63 | - | **41.47** | 62.81 |
| | TEA | 76.01 | 53.40 | - | 33.30 | 54.24 |
| | CRETTA | **83.65** | **68.86** | - | 40.04 | **64.18** |
| Sketch | Source | 13.89 | 14.50 | 19.03 | - | 15.81 |
| | TENT | **31.88** | **30.99** | 49.60 | - | **37.49** |
| | TEA | 18.60 | 25.47 | 48.34 | - | 30.80 |
| | CRETTA | 26.63 | 30.19 | **51.35** | - | 36.06 |

Table 3: ECE (↓) on each source domain of PACS.

| Method | P | A | C | S | AVG |
|---|---|---|---|---|---|
| TENT | 44.41 | 35.20 | 34.69 | 58.63 | 43.23 |
| TEA | 41.71 | 34.62 | 35.02 | **50.26** | 40.40 |
| CRETTA | **37.42** | **28.22** | **26.65** | 51.68 | **35.99** |

**Computational Efficiency** A major challenge for previous energy-based TTA methods was the high computational cost for SGLD sampling. This makes them impractical for real-time TTA scenarios that demand rapid adaptation. This computational burden becomes even more pronounced as the input sample size increases. More precisely, TEA not only incurs approximately six times the computational cost (213K GFLOPs) compared to CRETTA (34K GFLOPs) but also struggles to maintain competitive performance. In contrast, CRETTA enables adaptation without explicitly tracking the normalization constant within a pair-wise contrastive learning framework. As shown in Figure 2, CRETTA consistently outperforms comparison methods, including TENT and BN-Adapt, while maintaining relatively low GFLOPs, demonstrating its efficiency for real-time TTA.

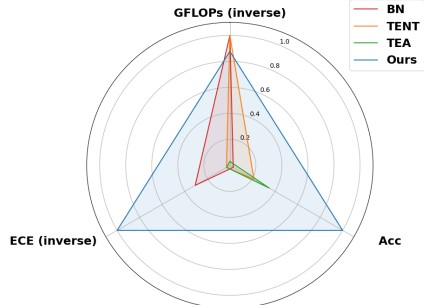

Figure 2: Comparison of GFLOPs, ECE and Acc against competitive baselines on TinyImageNet-C at the average across severity levels 1-5.

### 4.3 RESISTANCE MECHANISM TO OVERFITTING

In this subsection, we further analyze how the residual interpretation on distribution shift introduced in section 3 inherently provides a mechanism to mitigate overfitting and catastrophic forgetting.

**Performance Under Gradual Distribution Shift** To validate this mechanism, we conducted an experiment under a gradual distribution shift scenario. In this setting, the model continuously adapts from the source distribution $Q$ through increasing shift intensities $(1 \rightarrow 5)$, where severity 5 corresponds to the final target distribution $P$. After the model had fully adapted to $P$, we further froze the model and evaluated its classification accuracy on the original source distribution to observe if the target model remembers the original source distribution.

As summarized in Table 4, CRETTA demonstrates robust adaptation to progressively diverging target distributions,

Table 4: Gradual distribution shift on CIFAR10(-C) and CIFAR100(-C).

| Domain | CIFAR10 | | CIFAR100 | |
|---|---|---|---|---|
| | OURS | TEA | OURS | TEA |
| Source ($Q$) | **93.46** | 93.45 | **73.97** | 73.88 |
| 1 | **92.88** | 92.80 | **71.90** | 71.41 |
| 2 | **92.03** | 91.92 | **71.57** | 70.40 |
| 3 | **91.63** | 91.29 | **69.99** | 67.71 |
| 4 | **90.25** | 89.81 | **67.99** | 65.23 |
| 5 ($P$) | **89.47** | 88.78 | **65.47** | 60.26 |
| Source ($Q$) | **94.03** | 93.58 | **75.70** | 69.25 |

outperforming TEA with classification accuracy on both datasets. Notably, the model's accuracy on $Q$ improved after adaptation to $P$ (+1.73%). This improvement provides empirical evidence that CRETTA's residual-energy formulation acts as a regularizer that prevents forgetting and facilitates robust adaptation. In contrast, MLE-based method TEA lacks this regularization mechanism and sometimes suffers from forgetting, with performance degradation (-4.63%) after adaptation.

**Benefits of Contrastive Residual Energy Learning**  CRETTA tends to reduce target-sample energy during adaptation across severities as shown in Figure 3, enhancing robustness to strong distribution shifts (Yuan et al., 2024). Notably, CRETTA achieves this with up to **8× reduction of the computational cost** compared to existing energy-based method TEA.

While the observed energy reduction is meaningful, the core strengths of CRETTA lie in *contrastive residual energy learning* to prevent convergence to trivial solutions as discussed in section 3. The bias terms with respect to $E_\phi$ in Equation 3 prevent $E_\theta$ from becoming excessively small or large, thereby stabilizing the adaptation process. As shown in Figure 3, the energy of target samples increases drastically after adaptation without bias terms, resulting in performance degradation.

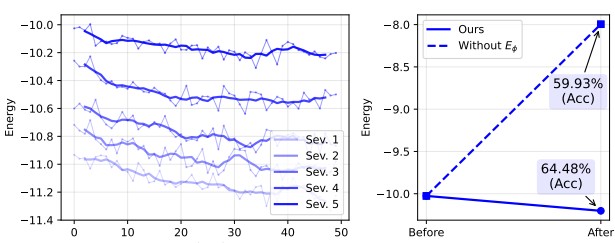

Figure 3: Energy trajectories of target samples across severities (left) and the effect of relative residual energy learning on stable adaptation (right) in CIFAR100-C.

Consequently, optimization proceeds adaptively based on the relative energy difference between source and target samples. When the energy difference between source and target is already aligned with the desired preference (i.e, $E_\theta(x_t) < E_\theta(x_s)$), the gradient weight $w(x_t, x_s)$ in Equation 4 decreases, leading to weaker updates. On the other hand, when the energy difference exists in the opposite direction of the target preference (i.e., $E_\theta(x_t) > E_\theta(x_s)$), the gradient weight $w(x_t, x_s)$ increases to enforce stronger corrections. By letting gradient updates depend on these relative energy differences, CRETTA automatically modulates learning strengths through the weighting term, enabling dynamic and stable adaptation.

### 4.4 ABLATION STUDIES

Other critical concerns regarding the replay buffer can be summarized in two folds: (1) Can the source data in the replay buffer be replaced with samples unseen during the pretraining phase? and (2) Does the model maintain its performance regardless of the quality of the samples included in the buffer?

Table 5: Effectiveness of buffer size.

| Buffer Ratio | CIFAR10-C | CIFAR100-C | TinyImageNet-C |
|---|---|---|---|
| 1% | 88.00% | 64.21% | 40.18% |
| 2% | 88.17% | 64.42% | 40.24% |
| 10% | 88.30% | 64.52% | 40.30% |

The first question becomes particularly important from a data privacy perspective when the original source data is unavailable. To address this, we evaluate the performance of CRETTA on CIFAR10-C with the replay buffer composed of CIFAR100 that we assume is distributionally similar but was not seen during the pretraining phase. As shown in Table 6, no performance drop is observed when the buffer is

Table 6: Effectiveness of source buffer content on CIFAR10-C.

| Buffer Type | Sev 5 | Sev 1-5 Avg. |
|---|---|---|
| Default(Random) | 88.30% | 91.01% |
| CIFAR100-trainset | 88.20% | 91.02% |
| CIFAR100-valset | 88.21% | 91.03% |

constructed from either CIFAR100 training or validation set. These findings suggest that CRETTA can operate effectively even without access to the original source data.

We further examined the performance of CRETTA in extreme cases where the source buffer composition is biased. Specifically, the buffer was constructed using high confidence (top 10%) and low confidence (bottom 10%) samples respectively based on the source model's confidence score. The results summarized in Table 7, indicate that adaptation performance remains unaffected by such variation in buffer content.

These comprehensive findings highlight that a small and randomly sampled buffer suffices for effective adaptation, regardless of its size or composition. This insensitivity to buffer configuration underscores the practicality of CRETTA, enabling deployment in real-world scenarios with minimal memory overhead and flexible buffer sourcing.

Table 7: Effectiveness of source buffer confidence.

| Buffer Type | CIFAR10-C | CIFAR100-C | TinyImageNet-C |
|---|---|---|---|
| Default (Random) | 88.30% | 64.52% | 40.30% |
| High Confidence | 86.92% | 64.10% | 40.18% |
| Low Confidence | 88.02% | 64.65% | 40.92% |

## 5 RELATED WORKS

**Test-time Adaptation**  TTA is an emerging paradigm that has demonstrated immense potential in adapting pretrained models to unlabeled test data during testing phase. Early methods such as batch normalization adaptations (BN Adapt) (Schneider et al., 2020) leveraged test-batch statistics, while techniques like TTT (Sun et al., 2020) and TTT++ (Liu et al., 2021) utilized image augmentations. TENT (Wang et al., 2020), minimizes entropy to update BN layers, aiming for improved adaptation but often resulting in overconfident 'that impair model calibration. EATA (Niu et al., 2022) and SAR (Niu et al., 2023) incorporates instance selection to filter unreliable samples, preserving performance, especially in continual test settings.

**Energy-based Models**  Energy-based models (EBMs) are non-normalized probabilistic models widely used in classification and generative tasks (Grathwohl et al., 2019; Guo et al., 2023; Kim & Bengio, 2016). Energy provides a non-probabilistic scalar value capturing the density of the data distribution, making EBMs effective for capturing distribution shifts (Du & Mordatch, 2019). Due to this property, energy-based approaches are utilized in out-of-distribution (OOD) detection and unsupervised domain adaptation (Herath et al., 2023). Recent works such as AEA (Choi et al., 2025) and TEA (Yuan et al., 2024) extend energy to test-time adaptation scenario.

**Learning by Comparison**  Noise-Contrastive Estimation (NCE) (Gutmann & Hyvärinen, 2010)performs maximum-likelihood estimation through nonlinear logistic regression, distinguishing real data from artificially generated noise. Although it provides insightful ideas as an optimization strategy, NCE still relies on the normalization constant implicitly, which can be difficult to handle in practice. By contrast, pairwise comparison removes the need for this constant by using linear logistic regression. Methods such as RLHF  (Ouyang et al., 2022) and DPO (Rafailov et al., 2024) adopt this idea within autoregressive text-generation models to better align generated responses with human preferences.

## 6 CONCLUSION

In test-time adaptation, the entropy minimization objective often suffers from poor calibration due to the overconfidence problem, while existing energy-based methods incur significant computational overhead from extensive sampling to approximate the normalization constant for the marginal target distribution. In contrast, CRETTA achieves reliable and efficient adaptation by redefining the distribution shift with a residual energy function while optimizing a contrastive objective that avoids the sampling.

CRETTA provides two key benefits. First, it inherently mitigates overfitting by adaptively reweighting gradient signals based on relative energy differences, thereby ensuring stable adaptation. Second, by embedding the residual energy function into the contrastive objective, CRETTA eliminates the need for normalization constant approximation. Through comprehensive experiments and ablations CRETTA confirms that it bridges the gap between calibration-aware adaptation and practical feasibility, offering a scalable solution previously unattainable with conventional TTA frameworks.

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

## A  TECHNICAL APPENDICES

### A.1  DERIVATION AND FUNCTION OF CRETTA

#### A.1.1  DERIVATION OF CRETTA

The marginal distribution of target data $p_\theta$ can be written as the product of the pretrained source model $q_\phi$ and an exponential residual term:

$$p_\theta(x) = \frac{1}{Z} q_\phi(x) \exp(-\frac{1}{\beta} \tilde{E}_\theta(x)),$$

where $\tilde{E}_\theta$ represents the residual energy function encoding the distribution shift. During TTA, the source model $q_\phi$ remains fixed, and the objective is to learn $\tilde{E}_\theta$ so as to align the source model more closely with the target distribution. By expanding the equation with respect to the energy function $\tilde{E}_\theta$, we can compute the residual energy score of an image sample $x$.

$$\tilde{E}_\theta(x) = -\beta \left( \log \frac{p_\theta(x)}{q_\phi(x)} + \log Z \right).$$

Next, we substitute the ground-truth residual energy function $\tilde{E}_\theta^*$ into the Bradley-Terry (BT) model (Bradley & Terry, 1952), which only depends on the difference in energy values between source and target pairs:

$$P(\tilde{E}(x_t) < \tilde{E}(x_s)) = \frac{1}{1 + \exp(-\tilde{E}_\theta(x_s) + \tilde{E}_\theta(x_t))} = \frac{1}{1 + \exp\left(\beta \log \frac{p_\theta(x_s)}{q_\phi(x_s)} - \beta \log \frac{p_\theta(x_t)}{q_\phi(x_t)}\right)},$$

where $x_t$ and $x_s$ denote the target and source samples, respectively. Here, for pairwise comparison, we use the negative residual energy.

Having derived the probability of the target distribution data in terms of the optimal energy function, which can further be expressed using $\phi$ and $\theta$, our objective for the target model is as follows:

$$\mathcal{L}(\theta; \phi) = -\mathbb{E}_{(x_s, x_t) \sim B} \left[ \log \sigma \left( \beta \log \frac{p_\theta(x_t)}{q_\phi(x_t)} - \beta \log \frac{p_\theta(x_s)}{q_\phi(x_s)} \right) \right] \tag{5}$$

In section 3, we emphasize that the key advantage of CRETTA is that it avoids the costly Stochastic Gradient Langevin Dynamics (SGLD) sampling required to compute the normalization constant as required in TEA (Yuan et al., 2024). However, the objective Equation 5 still includes the normalization constant for both target and source model.

To eliminate both normalization constants, we first redefine the target and source models using the Gibbs distribution as follows:

$$p_\theta(x) = \frac{\exp(-E_\theta(x))}{Z(\theta)} , q_\phi(x) = \frac{\exp(-E_\phi(x))}{Z(\phi)}$$

By integrating $p_\theta$ and $q_\phi$ into the Equation 5 and applying the logarithm, the normalization constants for both target and source model, i.e., $Z(\theta)$ and $Z(\phi)$, are canceled out as shown in below:

$$\mathcal{L}(\theta; \phi) = -\mathbb{E}_{(x_s, x_t) \sim B} \Big[ \ln \sigma \Big( \beta \big( -E_\theta(x_t) - \ln Z(\theta) + E_\phi(x_t) + \ln Z(\phi) \big)$$

$$- \beta \big( -E_\theta(x_s) - \ln Z(\theta) + E_\phi(x_s) + \ln Z(\phi) \big) \Big) \Big] \tag{6}$$

Therefore, the final learning objective is expressed as follows:

$$\mathcal{L}(\theta; \phi) = -\mathbb{E}_{(x_s, x_t) \sim B} \left[ \ln \sigma \left( \beta \left( -E_\theta(x_t) + E_\theta(x_s) + E_\phi(x_t) - E_\phi(x_s) \right) \right) \right]$$

### A.1.2 Function of CRETTA

In this section, we provide a detailed explanation of how each component of CRETTA contributes to adaptation, as well as the expected behavior during early and late stages of online adaptation.

If the target model $\theta$ successfully optimizes this objective, then residual energy function $\tilde{E}_\theta(x)$ in $p_\theta(x) = \frac{1}{Z} q_\phi(x) \exp(-\frac{1}{\beta} \tilde{E}_\theta(x))$ models the residual component of the distribution shift between the source and target domains.

At the beginning of adaptation, the model has not yet encoded the distribution shift. Therefore, the residual energy $E_\theta(x)$ is close to zero for both source samples $x_s$ and target samples $x_t$. This results in: $p_\theta(x) \approx q_\phi(x)$ meaning that predictions for both source and target data remain similar to the source model outputs.

As training progresses, the residual energy function learns the discrepancy between target and source distributions. For source samples $x_s$, $\tilde{E}_\theta(x_s)$ remains small, leading to $p_\theta(x_s) \approx q_\phi(x_s)$, preserving source performance. For target samples $x_t$, the residual energy adjusts predictions reflecting the learned domain shift and improving performance on the target domain. By progressively learning the residual while maintaining alignment with the source model, CRETTA achieves better generalization.

### A.1.3 Buffer Management of CRETTA

CRETTA initializes the source buffer $\mathcal{B}_s$ at model initialized, prior to adaptation, by randomly sampling source data up to a fixed buffer size, with an equal number of samples per class. During adaptation, the samples in the buffer are used sequentially in batches without any additional sampling or refresh, unlike TEA, thereby incurring no additional computational overhead.

Table 8: Comparison of classification accuracy (Acc ↑) and expected calibration error (ECE ↓) on the CIFAR10-C, CIFAR100-C, and TinyImageNet-C datasets at corruption severity level 5, the average across severity levels 1-5, and on clean data. The best adaptation results are emphasized in **BOLD**, while the second-best results are UNDERLINED.

| | CIFAR-10-C | | | | | CIFAR-100-C | | | | | TinyImageNet-C | | | | |
| | Clean | Corr Severity 5 | | Corr Severity 1-5 Avg | | Clean | Corr Severity 5 | | Corr Severity 1-5 Avg | | Clean | Corr Severity 5 | | Corr Severity 1-5 Avg | |
| Method | Acc(↑) | Acc(↑) | ECE(↓) | Acc(↑) | ECE(↓) | Acc(↑) | Acc(↑) | ECE(↓) | Acc(↑) | ECE(↓) | Acc(↑) | Acc(↑) | ECE(↓) | Acc(↑) | ECE(↓) |
|---|---|---|---|---|---|---|---|---|---|---|---|---|---|---|---|
| Source | 95.08% | 81.73% | 10.18% | 88.82% | 5.45% | 76.28% | 53.25% | 17.71% | 64.11% | 11.73% | 59.60% | 35.12% | 16.17% | 43.16% | 13.46% |
| *Normalization* | | | | | | | | | | | | | | | |
| BN Adapt | 93.59% | 85.46% | 4.85% | 89.12% | 3.15% | 72.84% | 60.74% | 8.32% | 65.83% | 6.88% | 56.72% | 39.60% | 13.66% | 44.72% | 12.12% |
| *Pseudo Labeling* | | | | | | | | | | | | | | | |
| PL | **94.85%** | 84.85% | 10.10% | 90.09% | 6.20% | **75.98%** | 56.33% | 23.81% | 65.72% | 16.66% | **58.95%** | 35.40% | 30.95% | 43.79% | 23.47% |
| SHOT | 94.38% | 87.91% | 5.42% | 90.78% | 3.86% | 75.00% | 64.41% | 8.93% | 68.80% | 7.44% | 56.90% | 39.84% | 13.81% | 44.95% | 12.24% |
| *Entropy Minimization* | | | | | | | | | | | | | | | |
| TENT | 94.35% | 87.84% | 5.49 % | 90.74% | 3.89 % | 74.95% | 64.31% | 8.93% | 68.73% | 7.47% | 56.92% | 39.83% | 13.82% | 44.94% | 12.24% |
| ETA | 93.72% | 85.46% | 4.85 % | 89.12% | 3.15 % | 73.71% | 61.77% | 8.54% | 66.66% | 7.10% | 56.82% | 39.67% | 13.70% | 44.79% | 12.16% |
| EATA | 93.72% | 85.46% | 4.85% | 89.12% | 3.15% | 73.66% | 61.79% | 8.54% | 66.65% | 7.11% | 56.86% | 39.68% | 13.70% | 44.79% | 12.16% |
| SAR | 93.61% | 86.54% | 4.79% | 89.80% | 3.13% | 73.73% | 62.71% | 8.31% | 67.36% | 6.91% | 56.77% | 39.66% | 13.72% | 44.77% | 12.16% |
| AEA | 94.21% | 88.27% | 5.09% | 90.88% | 3.73% | 75.17% | 64.40% | 9.16% | 68.75% | 7.61% | 56.97% | 39.87% | 13.82% | 44.97% | 12.25% |
| *Energy-based Models* | | | | | | | | | | | | | | | |
| TEA | 94.06% | 88.06% | **3.83%** | 90.67% | **2.68%** | 74.18% | 63.66% | **7.68%** | 67.93% | **6.33%** | 57.17% | 39.96% | 13.84% | 45.08% | 12.24% |
| CRETTA (Ours) | 94.43% | **88.30%** | 4.15% | **91.01%** | 2.88% | 75.26% | **64.52%** | 7.99% | **69.05%** | 6.82% | 58.23% | **40.30%** | **13.52%** | **45.75%** | **11.85%** |

## A.2 Additional Experiments and Analysis

### A.2.1 Detailed Performance Comparison

**Detailed Performance Comparison on Accuracy** Table 8 reports accuracy on the highest severity level 5, the average across severity levels (1-5), and performance on the clean dataset (i.e., without corruption) for CIFAR10-C, CIFAR100-C, and TinyImageNet-C. This table extends the results of Table 1 by additionally reporting accuracy on the clean dataset, providing a more complete view of model performance.

While CRETTA achieves the second-best accuracy on clean data among all methods, with the PL method performing the best. However, this can lead to overfitting and significant degradation in

Table 9: Comparison of expected calibration error (ECE ↓) on TinyImageNet-C datasets across all corruptions at the average across severity level 1-5. (Values are reported to one decimal place for space efficiency.)

| Method | Noise | | | Blur | | | | Weather | | | | Digital | | | | Avg |
|---|---|---|---|---|---|---|---|---|---|---|---|---|---|---|---|---|
| | Gauss. | Shot | Impul. | Defoc. | Glass | Motion | Zoom | Snow | Frost | Fog | Brit. | Contr. | Elastic | Pixel | JPEG | |
| Source | 12.5% | **11.6%** | 13.1% | 10.8% | 13.3% | 10.8% | 10.5% | 14.7% | 14.6% | 18.4% | 13.9% | 25.9% | 11.0% | **10.1%** | 10.6% | 13.5% |
| BN Adapt | 12.1% | 11.9% | **12.4%** | 11.0% | 11.9% | 11.0% | 10.3% | 13.1% | 12.7% | 14.0% | 12.1% | 16.7% | 10.9% | 10.7% | 10.9% | 12.1% |
| PL | 20.9% | 19.5% | 26.7% | 18.4% | 20.4% | 18.1% | 17.7% | 22.8% | 20.6% | 38.4% | 19.7% | 54.7% | 18.3% | 17.6% | 18.2% | 23.5% |
| SHOT | 12.2% | 12.1% | 12.5% | 11.1% | 12.1% | 11.2% | 10.5% | 13.2% | 12.8% | 14.2% | 12.2% | 16.9% | 11.0% | 10.7% | 11.0% | 12.2% |
| TENT | 12.2% | 12.1% | 12.5% | 11.1% | 12.1% | 11.1% | 10.5% | 13.2% | 12.8% | 14.2% | 12.2% | 16.9% | 11.0% | 10.8% | 11.0% | 12.2% |
| ETA | 12.1% | 12.0% | 12.5% | 11.1% | 12.0% | 11.1% | 10.4% | 13.1% | 12.7% | 14.1% | 12.2% | 16.7% | 10.9% | 10.7% | 10.9% | 12.2% |
| EATA | 12.1% | 12.0% | 12.4% | 11.1% | 12.0% | 11.1% | 10.4% | 13.2% | 12.7% | 14.1% | 12.2% | 16.8% | 10.9% | 10.7% | 10.9% | 12.2% |
| SAR | 12.1% | 12.0% | 12.5% | 11.1% | 12.0% | 11.1% | 10.4% | 13.2% | 12.7% | 14.1% | 12.2% | 16.8% | 10.9% | 10.7% | 10.9% | 12.2% |
| AEA | 12.2% | 12.1% | 12.5% | 11.2% | 12.1% | 11.2% | 10.4% | 13.3% | 12.8% | 14.2% | 12.2% | 16.9% | 11.0% | 10.8% | 11.0% | 12.3% |
| TEA | 12.1% | 12.0% | 12.6% | 11.2% | 12.1% | 11.1% | 10.5% | 13.2% | 12.7% | 14.1% | 12.2% | 16.9% | 11.0% | 10.8% | 11.0% | 12.2% |
| CRETTA | **12.0%** | 11.7% | 12.4% | **10.7%** | 11.9% | **10.5%** | **10.0%** | **12.7%** | **12.1%** | **13.6%** | **11.9%** | **16.6%** | **10.7%** | 10.3% | **10.6%** | **11.9%** |

performance under severe corruptions. Notably, while PL exhibits substantial drops in performance under corruption, CRETTA remains robust and effective across both clean and corrupted settings, demonstrating its reliability in both distributions.

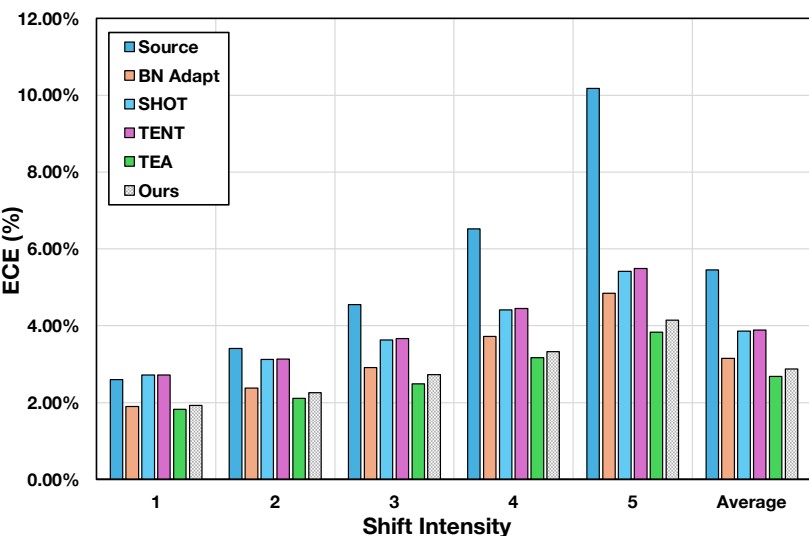

Figure 4: Comparison of Expected Calibration Error (ECE↓) on the CIFAR10-C dataset across different corruption severity levels.

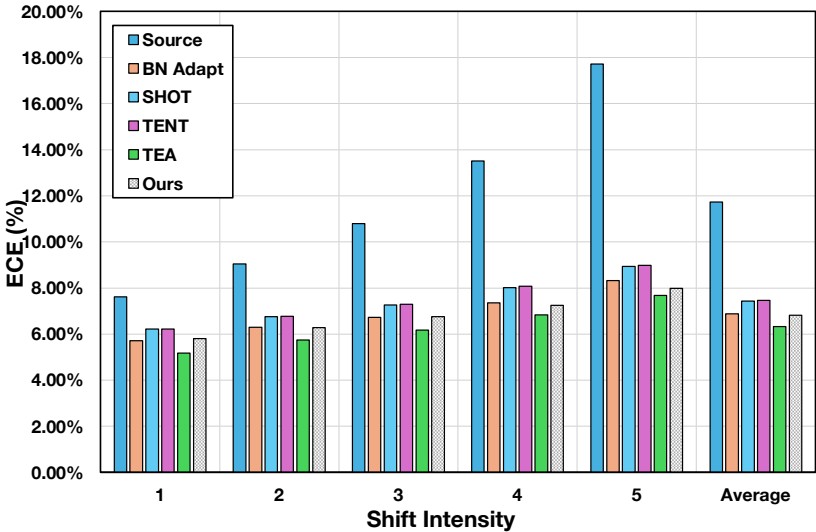

Figure 5: Comparison of Expected Calibration Error (ECE↓) on the CIFAR100-C dataset across different corruption severity levels.

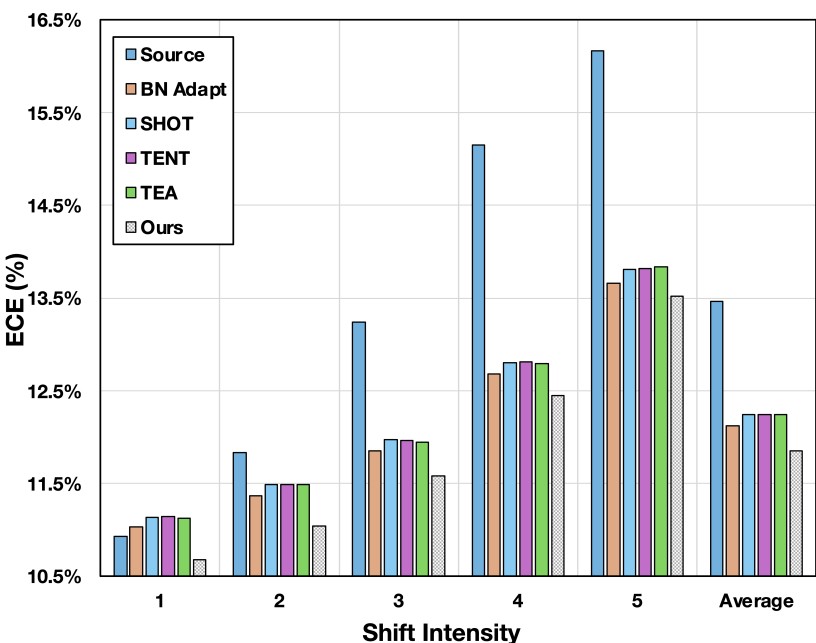

Figure 6: Comparison of Expected Calibration Error (ECE↓) on the TinyImageNet-C dataset across different corruption severity levels.

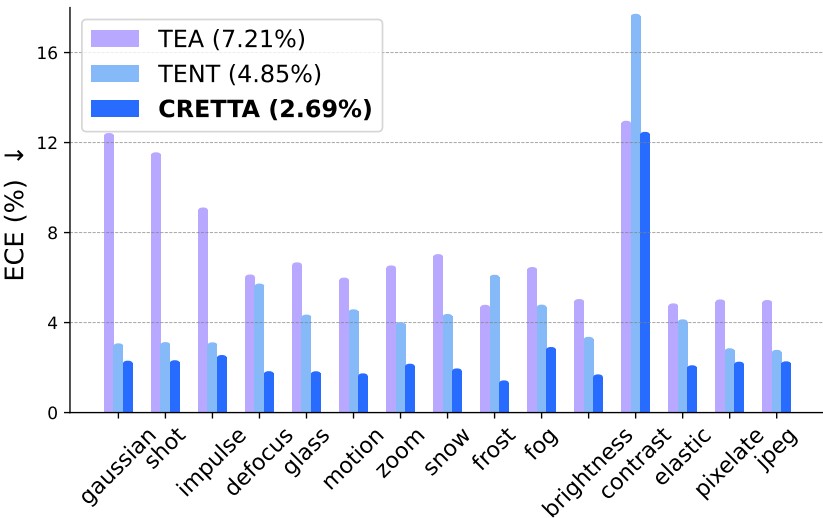

Figure 7: Comparison of Expected Comparison of Expected Calibration Error (ECE ↓) on ImageNet-C across various corruption types, with results averaged over severities 1–5..

**Detailed Performance Comparison on Calibration Error** In this section, we provide a detailed analysis of the Expected Calibration Error (ECE) for CIFAR10-C and CIFAR100-C. This expands upon the results shown in Table 1.

As seen in Figure 4 and Figure 5, energy-based methods such as TEA and CRETTA consistently outperform baseline approaches like TENT, which suffers from overconfidence issues. Furthermore, our method maintains computational advantage over TEA, making it more efficient while achieving comparable or superior performance.

On TinyImageNet-C dataset, shown in Figure 6, CRETTA outperforms all competing methods across all severity levels. This consistent superiority over all baseline methods demonstrates the robustness and adaptability of our approach in high-complexity datasets.

Table 10: Comparison of computational cost (GFLOPs), Memory Cost (Peak Memory Usage), and performance metrics (ECE and Acc) for baselines on the CIFAR10-C

|  | GFLOPs(↓) | Memory Cost(↓) | Acc(↑) | ECE(↓) |
|---|---|---|---|---|
| Source | 131.53 | 443.98 MB | 88.82% | 5.45% |
| BN Adapt | 131.53 | 452.61 MB | 89.12% | 3.15% |
| TENT | 132.59 | 1546.05 MB | 90.74% | 3.89% |
| TEA | 4335.82 | 3464.78 MB | 90.67% | 2.68% |
| Ours | 527.40 | 2651.83 MB | 91.01% | 2.88% |

Table 11: Comparison of computational cost (GFLOPs), Memory Cost (Peak Memory Usage), and performance metrics (ECE and Acc) for baselines on the CIFAR100-C

|  | GFLOPs(↓) | Memory Cost(↓) | Acc(↑) | ECE(↓) |
|---|---|---|---|---|
| Source | 131.53 | 443.03 MB | 64.11% | 11.73% |
| BN Adapt | 131.53 | 452.70 MB | 65.83% | 6.88% |
| TENT | 132.59 | 1546.21 MB | 68.73% | 7.47% |
| TEA | 4335.82 | 3465.00 MB | 67.93% | 6.33% |
| Ours | 527.40 | 2651.85 MB | 69.05% | 6.82% |

**Detailed Performance Comparison on Computational Efficiency** To further demonstrate the computational advantages of our proposed method, we present a comprehensive comparison of com-

Table 12: Comparison of classification accuracy (Acc ↑) and expected calibration error (ECE ↓) on the ImageNet-C dataset.

| Method | Severity L5 | | Severity Avg | |
|---|---|---|---|---|
| | Acc ↑ | ECE ↓ | Acc ↑ | ECE ↓ |
| TENT | **37.39%** | 7.75% | **43.78%** | 4.85% |
| TEA | 31.60% | 8.39% | 38.72% | 7.21% |
| CRETTA | 37.05% | **4.43%** | 43.54% | **2.69%** |

putational cost (GFLOPs), peak GPU memory usage (MB) with performance metrics (Accuracy and ECE) across CIFAR10-C, CIFAR100-C, and TinyImageNet-C, as summarized in Table 10, Table 11. Compared to TEA, which incurs substantial computational overhead due to SGLD-based sampling, CRETTA reduces GFLOPs by more than sevenfold across datasets. Furthermore, despite incorporating a source buffer, CRETTA maintains a modest peak GPU memory usage, significantly lower than TEA. The peak GPU memory usage is measured as the maximum allocated GPU memory during adaptation. Consequently, CRETTA offers a practical balance between performance, computational cost, and memory efficiency, making it well-suited for deployment in real-world, resource-constrained environments.

**Scalability**    In this section, we provide a detailed results on ImageNet-C.

As shown in Table 12, CRETTA achieves performance by a significant margin, outperforming the entropy-based method TENT and the existing energy-based method TEA in ECE.

Entropy minimizations's overconfidence and MLE-based approach's approximation error introduced when estimating its normalization constant term leads to poor calibration which is inappropriate in real-wold TTA scenarios. In contrast, CRETTA generalizes well to large-scale datasets, achieving strong predictive performance with superior calibration.

Table 13: Comparison of classification accuracy on CIFAR10(-C), CIFAR100(-C) under gradual distribution shift

| Domain | CIFAR10 | | | CIFAR100 | | |
|---|---|---|---|---|---|---|
| | OURS | TEA | TENT | OURS | TEA | TENT |
| Source (Q) | **93.46** | 93.45 | 93.43 | **73.97** | 73.88 | 73.57 |
| 1 | **92.88** | 92.80 | 92.77 | **71.90** | 71.41 | 71.70 |
| 2 | **92.03** | 91.92 | 91.92 | **71.57** | 70.40 | 71.36 |
| 3 | **91.63** | 91.29 | 91.35 | 69.99 | 67.71 | **70.04** |
| 4 | **90.25** | 89.81 | 90.03 | 67.99 | 65.23 | **68.28** |
| 5 (P) | **89.47** | 88.78 | 88.58 | **65.47** | 60.26 | 65.23 |

**Detailed Performance Comparison Under Gradual Shift scenario**    In subsection 4.3, we demonstrated that our contrastive residual energy-based learning shows superior performance over CD MLE-based adaptation method TEA. This tendency was consistently observed under the gradual distribution shift setting in Table 13, and here we additionally report comparisons with TENT.

For CIFAR10-C, CRETTA maintains the best performance throughtout the shift. For CIFAR100-C, CRETTA shows clear gains under stronger shifts. At severity 5, it achieves 65.47%, notably higher than TEA(60.26%) and TENT (65.23%). While TENT is compertitive at mid-level severities, it degrades more under severe shifts. Overall, CRETTA provides robust adaptation across gradual shifts while preventing forgetting, outperforming both TEA and TENT.

**Test-time Adaptation for Non-IID Settings**
Our previous experiments are conducted under the assumption of i.i.d. test samples which is a widely adopted setting in prior work. Nonetheless, real-world applications can also encounter non-i.i.d. samples (Gong et al., 2022; Yuan et al., 2023; Wang et al., 2022). To further examine the robustness and generalizability of our method beyond the i.i.d. assumption, we constructed a non-i.i.d. test-time adaptation scenario. Specifically, we simulated non i.i.d. data stream by leveraging a Dirichlet distribution to control the class allocation ratio within batch, denoted as $\delta$. A higher $\delta$ value brings the distribution closer to i.i.d., whereas a lower $\delta$ value results in a more non-i.i.d. distribution, where a specific class might dominate the batch. We conducted our experiment on CIFAR100-C using the WRN-28-10 backbone.

Table 14: Test-time adaptation in dynamic scenarios using CIFAR100-C at severity 5. Our method demonstrates higher robustness compared to baselines across varying the allocation ratio $\delta$.

| Method | $\delta = 10$ | $\delta = 1$ | $\delta = 0.1$ | $\delta = 0.01$ | Avg Acc. |
|---|---|---|---|---|---|
| BN Adapt | 61.44% | 61.11% | 59.02% | 45.61% | 56.79% |
| PL | 44.03% | 37.27% | 39.06% | 43.38% | 40.93% |
| SHOT | 63.94% | 63.60% | 61.20% | 46.54% | 58.82% |
| TENT | 63.91% | 63.56% | 61.20% | 46.72% | 58.85% |
| ETA | 62.31% | 62.04% | 59.89% | 46.04% | 57.57% |
| EATA | 62.35% | 62.04% | 59.84% | 46.04% | 57.57% |
| SAR | 61.54% | 61.22% | 59.12% | 45.66% | 56.89% |
| TEA | 62.58% | 62.29% | 60.08% | 46.22% | 57.79% |
| Ours | **66.20%** | **65.95%** | **63.47%** | **48.33%** | **60.99%** |

As Table 14 shows, our method consistently outperforms entropy minimization and instance selection approaches across all $\delta$ values. Specifically, CRETTA achieves the highest average accuracy of 60.99%, surpassing TENT's 58.85% by 2.14%p. Also, even at the most imbalanced setting where $\delta = 0.01$, our method achieves a competitive accuracy of 48.33%. These findings demonstrate that our method not only excels in i.i.d. scenarios but also is effective in dynamic real-world environments.

## A.3 ABLATION STUDY

### A.3.1 DETAILED ABLATION STUDY

Table 15: Comparison of classification accuracy(Acc) and expected calibration error(ECE) on benchmark datasets between CRETTA(Default) and CRETTA(Loss Term without Source Model) at severity level 5.

| Method | CIFAR10-C | | CIFAR100-C | | TinyImageNet-C | |
|---|---|---|---|---|---|---|
| | Acc($\uparrow$) | ECE($\downarrow$) | Acc($\uparrow$) | ECE($\downarrow$) | Acc($\uparrow$) | ECE($\downarrow$) |
| CRETTA | **88.30** | **4.15** | **64.52** | 7.99 | **40.30** | **13.52** |
| w.o Source Model Term | 88.09 | 4.66 | 60.02 | **5.93** | 37.46 | 14.55 |

**Loss Ablation** We observed that eliminating the source model consistently degraded both accuracy and calibration (ECE) in most cases across our benchmark datasets. These results collectively demonstrate that incorporating source model related terms into our contrastive residual learning is essential for stable adaptation.

**Gradient Ablation** The gradient coeffcient $w(x_t, x_s)$ is the key mechanism that turns relative energy into stable updates. To verify this role, we conducted an ablation study that disrupts the proposed weighting scheme by replacing $w(x_t, x_s)$ with values randomly sampled from a uniform distribution $[0, 1)$. As shown in Table 16, this replacement lead to lower accuracy and higher calibration error, confirming that gradient coefficient is critical for stable optimization and robust adaptation under noisy target data.

Table 16: Effect of Gradient Coefficient

| Method | CIFAR10-C | | CIFAR100-C | | TinyImageNet-C | |
|---|---|---|---|---|---|---|
| | Acc | ECE | Acc | ECE | Acc | ECE |
| Ours | **88.30** | 4.15 | **64.52** | 7.99 | **40.30** | 13.52 |
| Uniform | 87.47 | **4.13** | 61.66 | 8.03 | 38.33 | 15.13 |

Table 17: Comparison of classification accuracy(Acc) and expected calibration error(ECE) on benchmark datasets between CRETTA(Default) and CRETTA(Single Source Sample in Buffer) at severity level 5.

| Method | CIFAR10-C | | CIFAR100-C | | TinyImageNet-C | |
|---|---|---|---|---|---|---|
| | Acc(↑) | ECE(↓) | Acc(↑) | ECE(↓) | Acc(↑) | ECE(↓) |
| CRETTA | **88.30** | **4.15** | **64.52** | **7.99** | **40.30** | **13.52** |
| CRETTA with single source sample | 87.62 | 5.39 | 62.67 | 8.93 | **40.30** | 14.13 |

**Extended Buffer Ablation**   While the specific content of the buffer has less impact on performance, as shown in Table 5, this does not imply that the source buffer itself plays a trivial role. To further verify this, we additionally conducted an experiment where the buffer consists of only a single source sample. As shown in Table 17, accuracy dropped by up to 1.7% and ECE increased by up to 1.2% across datasets.

Table 18: Effectiveness of preference pair size on CIFAR10-C, CIFAR100-C, and TinyImageNet-C.

| | CIFAR10-C | CIFAR100-C | TinyImageNet-C |
|---|---|---|---|
| CRETTA wo/ CP | 88.30% | 64.52% | 40.30% |
| CRETTA w/ CP | 88.24% | 64.69% | 40.44% |

**Pair Size Ablation**   In CRETTA, we assume that the samples in a test batch represent the target distribution, while the source replay buffer represents the source distribution. The loss is computed by forming pairs between target and source samples within each batch, enabling a direct comparison between the two distributions.

To demonstrate the assumption is valid, we examined the impact of increasing the number of pair combinations using a Cartesian Product (CP) to generate all possible combinations of target and source data within each batch. For example, we use 200 pairs for each adaptation in CIFAR10-C, while the Cartesian Product results in 200×200 pairs.

Our results across three datasets summarized in Table 18 indicate that generating more pairs does not necessarily lead to performance gain. With only a few pairs, CRETTA can efficiently adapt to the target distribution.

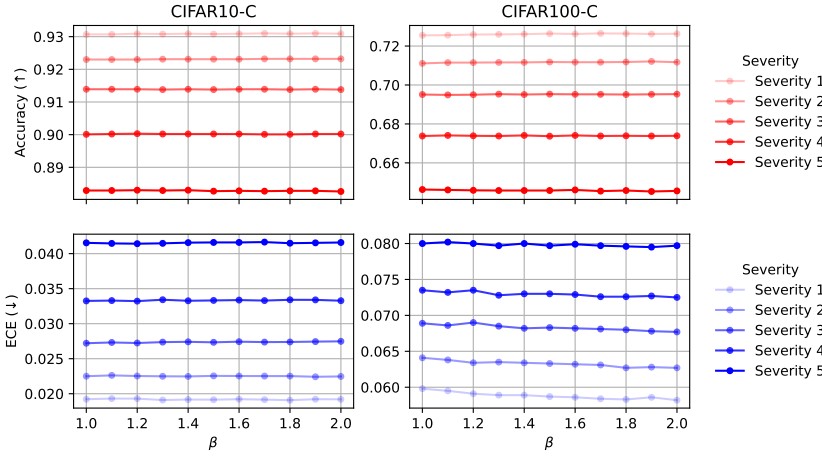

Figure 8: Ablation on varying $\beta$ values on CIFAR10-C and CIFAR100-C at severity 1-5.

**Hyperparameter $\beta$ Ablation**    The hyperparameter $\beta$ in Equation 3 controls the deviation from the pretrained source model, serving as a scaling parameter. To evaluate the robustness of our method, we experiment its performance across varying values of $\beta$, assessing both accuracy and expected calibration error (ECE) on CIFAR10-C, CIFAR100-C and TinyImageNet-C. As shown in Figure 8, our method consistently demonstrates stable performance across all corruption severity levels (1-5), validating its robustness.

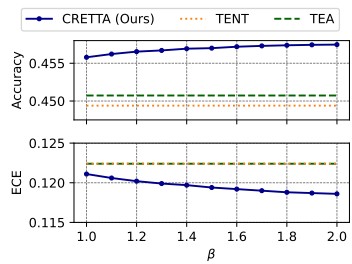

Figure 9: Ablation on varying values of $\beta$.

In addition, we further examine the effectiveness of CRETTA across varying values of the hyperparameter $\beta$ on TinyImageNet-C, averaging results over severity levels 1 to 5, and compare its performance against competitive baselines (see Figure 9). These results confirm that the strong adaptation performance of CRETTA is not reliant on a specific setting of the temperature parameter $\beta$, but rather stems from our contrastive residual learning objective itself.

### A.3.2    DETAILED SETTING OF CRETTA

Table 19: Detailed hyperparameters settings for each dataset.

| Dataset | LR | $\beta$ | Batch Size | Transformation Type (probability) |
|---|---|---|---|---|
| CIFAR10-C | 1e-3 | 1.0 | 200 | rotate(1.0) |
| CIFAR100-C | 2e-3 | 2.0 | 200 | flip, rotate, affine, perspective, crop(0.2) |
| TinyImageNet-C | 1e-3 | 2.0 | 1000 | None |

**Hyperparameters**    This section details the hyperparameter settings for CRETTA. To optimize performance, minimal hyperparameter tuning was conducted, focusing solely on learning rate, $\beta$ and type and probability of random transformations for source buffer. With only slight adjustments, CRETTA achieved significantly better performance than the current state-of-the-art (SOTA). The batch sizes were aligned with the default settings used in TENT and TEA, which are 200 for CIFAR10-C and CIFAR100-C, 1000 for TinyImageNet-C. For ImageNet-C we follow TENT default settings, using a batch size of 64 and learning rate of $2.5\mathrm{e}{-4}$. These settings ensured consistency across experiments while highlighting the robustness and effectiveness of CRETTA. For the PACS domain-generalization task, we used a learning rate of $1\mathrm{e}{-3}$, a batch size of 100, applying source-sample augmentation in the same way as for CIFAR100-C. All experiments were conducted using a single NVIDIA RTX A6000 GPU (48GB).

**Evaluation Metrics**    Expected Calibration Error (ECE) (Guo et al., 2017) is a metric used to measure the calibration quality of a probabilistic model. Calibration refers to how closely the predicted probabilities of a model match the actual probabilities. ECE quantifies the discrepancy between predicted confidence and actual accuracy. ECE is calculated as shown in Equation 7:

$$\mathrm{ECE} = \sum_{m=1}^{M} \frac{|\mathrm{bin}_m|}{N} \cdot |\mathrm{confidence}_m - \mathrm{accuracy}_m| \tag{7}$$

where $M$ is the number of bins, $N$ is the total number of data points, $\mathrm{bin}_m$ is the number of predictions in $m$-th bin, and $\mathrm{confidence}_m$ and $\mathrm{accuracy}_m$ are the confidence and accuracy of bin $m$, respectively.

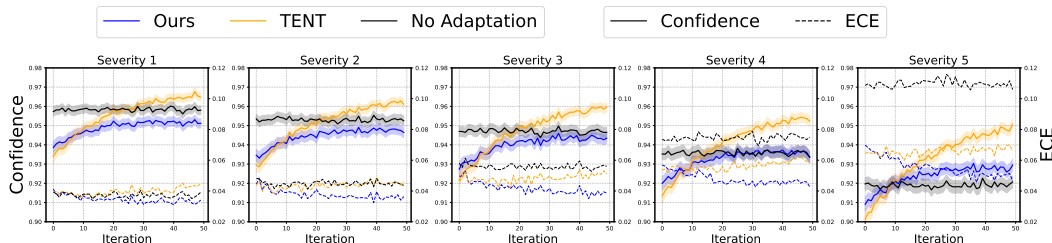

Figure 10: The overconfidence problem of entropy minimization in test-time adaptation on CIFAR10-C. TENT tends to increase a model's confidence in uncertain predictions as adaptation progresses, often leading to worse calibration due to overconfidence. In contrast, CRETTA (Ours) stabilizes the adaptation process by gradually reducing the expected calibration error.

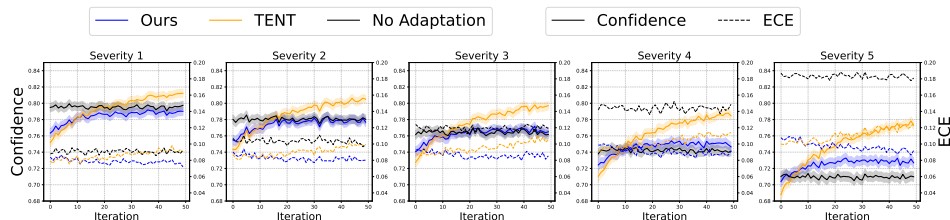

Figure 11: The overconfidence problem of entropy minimization in test-time adaptation on CIFAR100-C.

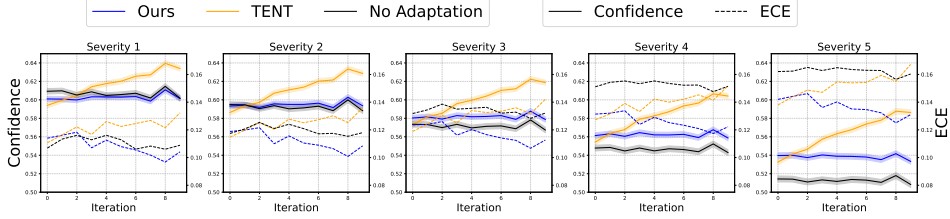

Figure 12: The overconfidence problem of entropy minimization in test-time adaptation on TinyImageNet-C.

### A.3.3 DETAILED RESULTS FOR OVERCONFIDENCE PROBLEM OF ENTROPY MINIMIZATION

The overconfidence issue inherent in entropy minimization has been thoroughly investigated in prior works (Liu et al., 2020; Hendrycks & Gimpel, 2016; Guo et al., 2017). Building on this, we explored that increasing a model's prediction confidence especially when the label information is unavailable can lead to bad calibration as shown in Figure 4. The trend consistently appears in other benchmark datasets including CIFAR100-C and TinyImageNet-C as illustrated in Figure 11 and Figure 12. Entropy minimization raises the model's confidence across all severity levels, with the rate of increase becoming steeper as corruption severity intensifies, thereby exacerbating error accumulation.

On the other hand, CRETTA maintains stable confidence managing uncertainty during test-time adaptation and even reduces calibration error as adaptation progresses. These results suggest that maximizing the marginal likelihood of target samples provides a safer and more effective strategy compared to relying on uncertain predicted probabilities $p_\theta(\hat{y}|x)$ in the test-time learning objective.

## B NOISE CONTRASTIVE ESTIMATION

We first define a reward function $r(\cdot)$ to properly compare samples from two different sets or distributions.

$$r(x; \theta, \phi) = \log P_\theta(x) - \log P_\phi(x)$$

where $P_\theta$ is the target distribution and $P_\phi$ is the source distribution.

## B.1 Non-residual

If we define energy functions for each of them by utilizing gibbs distribution,

$$E_\theta(x) = -\log P_\theta(x) - \log Z(\theta)$$
$$E_\phi(x) = -\log P_\phi(x) - \log Z(\phi)$$

Then the reward function becomes

$$r(x; \theta, \phi) = -(E_\theta(x) - E_\phi(x)) + C$$

Then the loss function becomes

$$\mathcal{L}(\theta; \phi) = -\mathbb{E}_{x_t}\left[\log \sigma(r(x; \theta, \phi))\right] - \mathbb{E}_{x_s}\left[\log(1 - \sigma(r(x; \theta, \phi)))\right]$$

## B.2 Residual

If we define a residual energy function,

$$p_\theta(x) = \frac{1}{Z} q_\phi(x) \exp(-\frac{1}{\beta}\tilde{E}_\theta(x))$$

Then the reward function becomes

$$r(x; \theta, \phi) = \log p_\theta(x) - \log q_\phi(x) = -\frac{1}{\beta}\tilde{E}_\theta(x) + c$$

Then the loss function becomes

$$\mathcal{L}(\theta; \phi) = -\mathbb{E}_{x_t}\left[\log \sigma(r(x; \theta, \phi))\right] - \mathbb{E}_{x_s}\left[\log(1 - \sigma(r(x; \theta, \phi)))\right]$$

# C Pair-wise Contrastive Estimation

We first define a reward function $r(\cdot)$ to properly compare samples from two different sets or distributions.

$$r(x_t, x_s) = \tilde{r}(x_t) - \tilde{r}(x_s)$$

where $\tilde{r}$ is a reward function that assigns higher values to target samples than source samples

## C.1 Non-residual

If we define energy functions for each of them,

$$E_\theta(x) = -\log P_\theta(x) - \log Z(\theta)$$

Then the reward function becomes

$$r(x_t, x_s) = \log P_\theta(x_t) - \log P_\theta(x_s) = -(E_\theta(x_t) - E_\theta(x_s))$$

Then the loss function becomes

$$\begin{aligned}\mathcal{L}(\theta; \phi) &= -\mathbb{E}_{x_t, x_s}\left[\log \sigma(r(x_t, x_s))\right] \\ &= -\mathbb{E}_{x_t, x_s}\left[\log \sigma(-(E_\theta(x_t) - E_\theta(x_s)))\right]\end{aligned}$$

The gradient becomes

$$\begin{aligned}\nabla_\theta \mathcal{L}(\theta; \phi) &= -\mathbb{E}_{x_t, x_s}\left[\sigma(-r(x_t, x_s))\nabla_\theta r(x_t, x_s)\right] \\ &= \mathbb{E}_{x_t}\left[\sigma(E_\theta(x_t) - E_\theta(x_s))\left(\nabla_\theta E_\theta(x_t) - \nabla_\theta E_\theta(x_s)\right)\right]\end{aligned}$$

## D    USE OF LLMs

We used a large language model (ChatGPT) only as a general purpose assistive tool for minor editing tasks such as polishing sentences, correcting grammar and spelling and making small LaTeX table formatting adjustments. The LLM was not involved in research ideation, experimental design, data analysis, or substantive writing. All technical decisions, interpretations, and the writing of the core content were carried out entirely by the authors, who take full responsibility for the originality of the manuscript.

## E    REBUTTAL

**Clarifying Why Energy Minimization Improves Generalization and How It Differs from Entropy Minimization**    In this section, we clarify (1) why energy minimization improves generalization capability and (2) how this approach differs, especially from an optimization perspective, from entropy minimization, which is known to suffer from overconfidence issues.

First, to understand why minimizing the energy of target samples leads to improved generalization capability (i.e., higher classification accuracy) on the target distribution, it is essential to examine how optimizing the energy function reshapes the representations.

We assume a classifier $f_\theta(x) = g(h_\theta(x))$ with feature extractor $h_\theta$ and a frozen linear classifier $g(z) = Wz + b$, where $W = [w_1, \ldots, w_C]^\top \in \mathbb{R}^{C \times d}$. The logits can be expressed as

$$a_k(x) = w_k^\top h_\theta(x) + b_k.$$

Since target samples contain no labels, we optimize the marginal energy of the input,

$$E_\theta(x) = -\log \sum_{k=1}^{C} \exp(a_k(x)),$$

which is the standard unnormalized negative log-density in energy-based models. The gradient of this energy with respect to the logit $a_k(x)$ is

$$\frac{\partial E_\theta(x)}{\partial a_k(x)} = -\frac{\exp(a_k(x))}{\sum_{k'=1}^{C} \exp(a_{k'}(x))} = -p_\theta(y = k \mid x).$$

Using $\frac{\partial a_k}{\partial z} = w_k$, the gradient of the energy with respect to the feature representation becomes

$$\frac{\partial E_\theta(x)}{\partial z} = \sum_{k=1}^{C} \frac{\partial E_\theta(x)}{\partial a_k} \frac{\partial a_k}{\partial z} = -\sum_{k=1}^{C} p_\theta(y = k \mid x) w_k = -\mathbb{E}_{p_\theta(y|x)}[w_y].$$

This expression shows that the gradient descent updates shift the feature $z$ toward the expected classifier weight vector. Although the classifier remains frozen, this shift alters the logits and thus reshapes the conditional distribution, enabling the model to align its predictions to the target domain solely through feature-level adjustments. From a representation perspective, energy minimization adapts the feature extractor to better capture the target-domain distribution, producing stronger representations that improve classification accuracy under distribution shift even without labels.

Furthermore, to understand how energy minimization and entropy minimization objectives behave differently during optimization, it is essential to compare their gradients. Since we already expressed the gradient of the energy objective, we now present the gradient expressions for the entropy objective.

For unlabeled target data, the entropy of model prediction is expressed as

$$\mathcal{L}_{\text{ent}}(x) = -\sum_{k=1}^{C} p_k \log p_k,$$

where $p_k = p_\theta(y = k \mid x)$. The corresponding gradient is

$$\frac{\partial \mathcal{L}_{\text{ent}}}{\partial z} = -\sum_{y=1}^{C} (\log p_y + 1) p_y \left( w_y - \sum_{k=1}^{C} p_k w_k \right).$$

Letting $\mathbb{E}[w] = \sum_k p_k w_k$ denote the expectation of classifier weights under the current predictive distribution, we obtain the final compact form:

$$\frac{\partial \mathcal{L}_{\text{ent}}}{\partial z} = -\sum_{y=1}^{C}(\log p_y + 1)p_y\,(w_y - \mathbb{E}[w])\,.$$

In the gradient, the term $(\log p_y + 1)p_y$ heavily weights classes for which $p_y$ is already large and $\log p_y$ is less negative. Thus, the gradient of entropy moves the feature $z$ in the direction that increases confidence for the most likely classes, effectively reducing prediction entropy and behaving similarly to pseudo-labeling. By directly modifying the conditional distribution, entropy minimization mainly pushes the model to become more confident, often excessively.

In contrast, the gradient of the energy objective depends only on $\mathbb{E}_{p_\theta(y|x)}[w_y]$, which is a smooth expectation over classifier weights. It does not contain the entropy term's confidence-amplifying multiplier. This trend is also empirically confirmed in Figure 10, Figure 11, and Figure 12.

Overall, energy minimization improves generalization by altering the logits and reshaping the conditional distribution, allowing the model to align its predictions to the target domain. While achieving strong classification performance, it is also more robust than entropy minimization because it optimizes a smooth expectation over classifier weights, avoiding the entropy objective's confidence-amplifying multiplier. Together, these properties make energy-based adaptation a more balanced and principled optimization approach, enhancing representation quality rather than merely increasing confidence, which explains its superior robustness in test-time adaptation.

**Extended Buffer Ablation** While the specific content of the buffer has less impact on performance, as shown in Table 5, this does not imply that the source buffer itself plays a trivial role. To further verify this, we additionally conducted an experiment where the buffer consists of only a single source sample. As shown in Table 17, accuracy dropped by up to 1.7% and ECE increased by up to 1.2% across datasets.

To understand why this is the case, recall that source energy (i.e., $E(x_s)$) is required not merely because of residual learning, but because it is a crucial component of the pairwise contrastive objective. Using an arbitrary reference energy as source energy would likely cause training failure. From a gradient perspective, Equation 4 shows that during early adaptation, a high source energy drives the gradient weight $w$ to collapse to zero, resulting in a trivial solution where learning cannot proceed—mirroring the well-known constraint in Noise Contrastive Estimation (NCE) regarding the choice of the noise distribution. To enable effective early adaptation, the buffer must therefore contain samples drawn from a distribution similar to that learned by the pretrained source model, ensuring that these samples receive low energy and provide meaningful contrastive learning signals.

Table 20: Performance Comparison of Source Buffer Contents on CIFAR100-C

| Buffer Type | Severity 5 | Severity 1–5 |
|---|---|---|
| **CRETTA (ours)** | **64.52%** | **69.05%** |
| CIFAR-10 (train) | 64.97% | 69.37% |
| CIFAR-10 (val) | 64.95% | 69.37% |
| PACS (sketch) | 63.74% | 68.25% |

This behavior is confirmed empirically. As shown in Table 20, CRETTA maintains strong performance on CIFAR100-C when the buffer contains CIFAR-10, which is not original source data but is distributionally similar. In contrast, performance deteriorates when the buffer contains samples with a very different distribution, such as PACS.

Thus, Table 6 should not be interpreted as suggesting that the absolute source-energy distribution plays a minor role. Rather, it demonstrates that CRETTA remains robust and consistently effective as long as the buffer contains data that are distributionally similar to the source distribution—even without access to the original source dataset. In practice, while true source data may be inaccessible due to privacy constraints, obtaining similar samples is usually far more feasible.

Overall, CRETTA's superior performance does not arise merely from lowering target energy; instead, it results from the interplay between the residual formulation, the pairwise contrastive objective, and their integration, which together provide a stable and powerful adaptation mechanism.

**Contrastive Component Ablation**    In this section, we clarify why the contrastive component is necessary and how it contributes to stable adaptation.

- **(1) Why — Contrastive learning is necessary for well-calibrated and stable adaptation.** We demonstrate this by *removing the contrastive term*, showing that direct minimization of target energy leads to unstable energy collapse and degraded calibration across benchmarks.

- **(2) How — Contrastive learning provides a more informative gradient signal.** We validate this by *replacing source-buffer samples with target samples*, showing that target-only regularization produces weak gradients and yields only marginal adaptation.

First, to empirically validate whether contrastive learning is indeed essential for stable adaptation, we first conducted an additional experiment in which we removed the contrastive term and measured the resulting calibration error across the three benchmark datasets.

Table 21: Ablations on the contrastive component across benchmark datasets (ECE).

| | CIFAR10-C Sev5 | CIFAR10-C Sev1–5 | CIFAR100-C Sev5 | CIFAR100-C Sev1–5 | TinyIN-C Sev5 | TinyIN-C Sev1–5 |
|---|---|---|---|---|---|---|
| **W Contrastive Terms (CRETTA)** | 4.15% | 2.88% | 7.99% | 6.82% | 13.52% | 11.85% |
| **W/O Contrastive Terms** | 5.57% | 4.08% | 11.61% | 9.65% | 16.21% | 14.12% |

As shown in Table 21, removing the contrastive terms (i.e., $E(x_s)$) and directly minimizing the target energy led to a consistent degradation in calibration across all benchmarks. The effect was particularly pronounced on CIFAR100-C, where the calibration error deteriorated to 11.61% (+4%p), which is notably worse than TENT (8.93%), an entropy minimization–based method that is prone to overconfidence. These results clearly demonstrate that the stability of CRETTA's adaptation does not arise simply from reducing target energies, but instead stems from the contrastive learning mechanism.

To further analyze how the contrastive term contributes to stable adaptation, we also examined the behavior of energy levels and calibration error on CIFAR100-C (Severity 5) during adaptation, comparing CRETTA against the variant where the contrastive terms are ablated.

Table 22: Target energy and ECE of CRETTA vs. without contrastive terms during adaptation on CIFAR100-C (Severity 5).

| | Target Energy | | ECE | |
|---|---|---|---|---|
| **Batch Idx** | **W Contrastive (OURS)** | **W/O Contrastive** | **W Contrastive (OURS)** | **W/O Contrastive** |
| 0 | -9.9781 | -9.9781 | 11.11% | 11.56% |
| 9 | -10.1208 | -10.9536 | 10.48% | 11.64% |
| 19 | -10.2124 | -11.5247 | 9.48% | 12.97% |
| 29 | -10.1896 | -11.7863 | 9.85% | 12.35% |
| 39 | -10.1856 | -12.0177 | 8.88% | 13.35% |
| 49 | -10.1904 | -12.2531 | 9.15% | 12.92% |
| $\Delta$ **(Last–First)** | **-0.21** | **-2.28** | **-1.96%** | **+1.35%** |

As shown in Table 22, we observe that when the contrastive terms are removed and the model directly minimizes the target energy, the energy level drops rapidly during the early stages of adaptation. While this may facilitate fast initial adaptation, it poses a critical risk to stability since aggressively lowering target energies early on sharpens the energy landscape around target samples, which can lead to overfitting. Empirically, we indeed find that removing the contrastive terms results in an overall increase in calibration error.

In contrast, CRETTA reduces the energy level progressively within the contrastive learning framework, enabling a more stable adaptation trajectory. This gradual reduction improves calibration error over time, demonstrating that the contrastive mechanism plays a key role in stabilizing adaptation dynamics.

Table 23: Gradient coefficient $w$ of CRETTA and Target-as-Source Buffer Data during adaptation on CIFAR100-C (Severity 5).

| Batch Idx | CRETTA | W TRG as SRC |
|---|---|---|
| 0 | 0.490 | 0.304 |
| 9 | 0.535 | 0.406 |
| 19 | 0.580 | 0.456 |
| 29 | 0.580 | 0.465 |
| 39 | 0.627 | 0.518 |
| 49 | 0.608 | 0.522 |
| **AVG** | **0.572** | **0.454** |

Table 24: Performance comparison of Target-as-Source buffer setting on CIFAR100-C (Severity 5).

| Method | Sev5 Acc | Sev5 ECE |
|---|---|---|
| BN Adapt | 60.74% | 8.32% |
| W TRG as SRC | 61.39% (+0.65%p) | 8.19% |
| **CRETTA** | **64.52% (+3.78%p)** | **7.99%** |

Furthermore, the contrastive learning mechanism proposed in CRETTA that utilizes a small buffer of source (or distributionally similar) samples provides a more meaningful gradient signal than using target samples for regularization. In our previous comment, we illustrated this using an extreme scenario where the source energy, driven by high-energy target samples, becomes sufficiently large that the gradient effectively vanishes. Here, we provide a more realistic explanation focusing on the relative magnitudes of the energy level differences.

Target samples within the same batch are drawn from the same underlying distribution, and so thus it is unlikely for their energy levels to differ significantly. In contrast, source (or distributionally similar) samples originate from distribution that the pretrained model has already learned, making them more likely to exhibit consistently lower energy values than newly encountered target samples. This creates a meaningful energy gap from the target sample energies, which in turn provides a strong gradient signal during adaptation. Consequently, CRETTA's contrastive learning framework yields notable gains in classification performance while achieving better calibration.

CRETTA provides a substantially more informative learning signal than simply replacing source samples with low-energy target samples. To validate this, we constructed an experimental setup where the 50% of samples with the lowest model-computed energy values in each target batch served as source-buffer data to compute the source energy $E(x_s)$. We then compared this setup with CRETTA's adaptation process and performance.

More concretely, we first compared the magnitude of the gradient coefficient $w$ throughout the adaptation process. As shown in Table 23, replacing source samples with low energy target samples results in consistently smaller gradient coefficients than CRETTA across the entire adaptation trajectory. This indicates that the model receives weaker learning signals and therefore fails to sufficiently adapt to the target distribution. Consequently, as shown in Table 24, the accuracy improvement is only marginal amounting to just +0.65 percentage points compared to BN adapt, which performs adaptation solely through normalization without any learning. In contrast, CRETTA maintains a relatively meaningful gradient coefficient while gradually increasing the learning signal from the early, high-uncertainty stages of adaptation toward later stages. This leads to both improved classification performance and better calibration, ultimately achieving effective and stable adaptation.

Overall, CRETTA's contrastive learning is an essential component for achieving well-calibrated and stable test-time adaptation. By progressively lowering target energy and thereby reducing calibration error, it enables a stable adaptation process. Moreover, CRETTA's contrastive learning methodology, which leverages buffer data, is distinctly more effective than approaches that apply only target-sample-based regularization. Notably, CRETTA's contrastive framework is also robust to variations in buffer-data content and quality, making it highly practical for real-world deployment.

