# OpenReview forum: "Contrastive Residual Energy Test-time Adaptation"
_ICLR.cc/2026/Conference — Submitted to ICLR 2026_

### Official Review · Reviewer_SoyK · 2025-10-14

**Soundness:** 3
**Presentation:** 3
**Contribution:** 2
**Rating:** 2
**Confidence:** 4

**Summary:**

The paper aims to improve model robustness under distribution shift without access to labeled source data. The authors argue that existing TTA methods relying on conditional distributions suffer from poor calibration, while energy-based approaches, though label-free, are computationally expensive due to sampling. To address these limitations, the authors propose Contrastive Residual Energy Test-Time Adaptation (CRETTA), which defines a residual energy function over target data and incorporates it into a contrastive objective. An adaptive gradient reweighting mechanism is used to mitigate overfitting and eliminate the need for sampling. Experimental results are reported to show that CRETTA achieves better calibration and efficiency compared to prior TTA methods.

**Strengths:**

1. Test-time adaptation remains an active and challenging area, and the paper’s focus on calibration and computational efficiency is well-motivated.

2. The integration of residual energy modeling with a contrastive objective is conceptually interesting and may open paths toward energy-efficient adaptation.

**Weaknesses:**

1. While the paper presents an interesting reformulation of energy-based adaptation, the overall contribution appears incremental—largely combining existing ideas (energy modeling, contrastive learning, and gradient reweighting) with limited novelty.

2. According to Table 1, the experiment improvements are marginal at best, and are worse in many cases. In general the empirical section demonstrates some improvements but does not convincingly establish robustness, scalability, or significant gains over strong baselines.

**Questions:**

1. Why does not the performance improve over TEA in Table 1? Why is the performance improvement is larger in Table 4?

2. What is the computational cost of CRETTA relative to energy-based methods that rely on sampling?

3. "CRETTA consistently outperforms other methods on most of corruption types in calibration as reported in Table 9." But there is no Table 9, there are only 7 tables.

---

> ### Author Response · Authors · 2025-11-21
>
> # Novelty of CRETTA
> ---
> We first thank the reviewer for acknowledging the strengths of CRETTA, particularly its efficiency and its pathway toward energy-efficient adaptation. We would also like to take this opportunity to further clarify the contributions of our approach.
>
> To the best of our knowledge, we are the **first to introduce a residual formulation for TTA** in the absence of ground-truth labels, whereby the adaptation task is redefined so that the model learns only the residual components that the pretrained model has not yet captured. Furthermore, this perspective also aligns with standard TTA practice that for stable adaptation, the backbone is typically frozen except for the BN layers and only a small subset of the model is updated. This small, selectively updated subset corresponds to the residual component adjusted at test time. Our formulation can thus be seen as extending this architectural design choice into an explicit learning-objective framework.
>
> Although the energy-based model (EBM) used in our framework may appear familiar, the key novelty of CRETTA does not lie in simply optimizing $p(x)$ utilizing an EBM. Instead, **our main contribution is that, within the residual learning framework, we integrate a contrastive learning objective that enables a simple mathematical trick to eliminate the normalization constant**, and devise a **computationally efficient yet reliable** form of test-time adaptation, one that existing adaptation strategies have not been able to achieve.
>
> Moreover, the effectiveness of our methodology is validated through extensive experiments: across datasets of varying scales (from CIFAR to ImageNet and PACS), diverse test-time settings (standard, gradual shift, non-iid and episodic), and comprehensive ablations (contrastive component–buffer, residual component–source model, temperature, etc.).
>
> We truly appreciate your thoughtful feedback and hope our responses help illuminate the value and contributions of our work.

---

> ### Author Response · Authors · 2025-11-21
>
> # W2&Q1: Clarifying the Scalability, Significance and Robustness of CRETTA
> We appreciate the reviewer’s concern regarding the perceived relatively marginal improvements in Table 1 and the stronger gains observed in Table 4. Below, we clarify the underlying reason for this behavior and how it aligns with the core contribution of CRETTA.
>
> - **Why Improvements Are Comparable in Table 1 (Small-scale)** : TEA’s approximation errors do not accumulate on small or static settings but grow as datasets scale or adaptation becomes dynamic, whereas CRETTA remains stable, achieving balanced accuracy, calibration, and efficiency on small datasets and meaningful, scalable gains on larger benchmarks.
>
> - **Why Performance Improvements Are Much Larger in Table 4 (Gradual Distribution Shift)** : Overall, CRETTA is a competitive method that delivers meaningful improvements over existing baselines while maintaining robustness under realistic distribution shifts, demonstrating its readiness for real-world TTA.
>
> ---
>
> ### **Why Improvements Are Comparable in Table 1 (Small-scale)**
>
> > Table 1 evaluates CIFAR-10/100-C, where the images are small (32x32) and semantically simple. In this constrained setting, **TEA’s sampling-based normalization errors do not yet accumulate, so its inherent instability does not appear.** Even so, CRETTA attains SOTA accuracy and 8x higher efficiency, with the second-best ECE despite removing sampling entirely. This reflects CRETTA’s design goal of balanced accuracy, calibration, and efficiency, which is essential for practical TTA. The small ECE gap is therefore expected and not a limitation.
> >
> > ----
> > - **Evidence from Larger Datasets Confirms the Significance and Scalability of CRETTA**
> >
> > **[Table A]** Performance Comparison of TEA and CRETTA (Ours) on ImageNet-C (%).
> | | Sev5 acc | Sev5 ece | Sev1-5 acc | Sev1-5 ece |
> | :--- | :--- | :--- | :--- | :--- |
> | **TEA** | 31.60 | 8.39 | 38.72 | 7.21 |
> | **CRETTA** | **37.05 (+5.45)** | **4.43 (-3.96)** | **43.54 (+4.82)** | **2.69 (-4.52)**|
> >
> >**On larger and semantically diverse datasets**, (TinyImageNet-C in Table 1, ImageNet-C in Table A), **TEA’s instability emerges, while CRETTA maintains stable improvements**:
> >TinyImageNet-C (64×64, 200 classes) and ImageNet-C (224×224, 1k classes) introduce much higher semantic complexity compared to CIFAR-C. While TEA appears stable on small datasets, it becomes increasingly unstable as dataset size and adaptation horizon grow. As shown in Figure 1, TEA’s repeated sampling and normalization estimation introduce accumulating approximation errors, leading to degraded calibration during adaptation in ImageNet-C. In contrast, **CRETTA consistently reduces calibration error and improves accuracy over time, maintaining stable performance gains**, which is further confirmed in another large scale dataset PACS with substantial style variation (Table 2 and Table 3).
>
> ---
>
> ### **Why Performance Improvements Are Much Larger in Table 4 (Gradual Distribution Shift)**
> >
> > Furthermore, CRETTA's strength becomes more evident in realistic, dynamically changing environments. As you noted, its performance gains in Table 4, which evaluates adaptation under a gradual and continuous distribution shift (Q → 1 → 2 → 3 → 4 → P) are significantly larger than in Table 1. **In this more realistic scenario, where the model must adapt sequentially as the corruption severity progressively increases, CRETTA maintains stable performance throughout the entire adaptation process**, extending reliably beyond the standard TTA setting. Its residual-energy formulation serves as an effective regularizer that prevents forgetting and enables robust adaptation. In contrast, TEA accumulates approximation errors over time and eventually suffers from compounding overfitting and catastrophic forgetting.
> >
> > ---
> > - **Evidence from Dynamic Settings Confirms the Robustness of CRETTA  (Table 14 in Appendix A.2)**
> >
> > CRETTA also performs reliably in non-IID streaming scenarios where test batches exhibit class imbalance and many baselines experience severe degradation. Under this setting, CRETTA achieves SOTA accuracy 60.99%, outperforming TENT (58.85%) and TEA (57.79%).
>
> ---
> ### **Conclusion**
>
> - The modest gains in Table 1 are expected because **TEA’s approximation errors** do not accumulate on small datasets and static scenario, masking its inherent instability.
> - As dataset size changes or adaptation scenario becomes dynamic, **these errors compound and degrade TEA’s performance, whereas CRETTA remains stable, resulting in substantially larger improvements on Table 4 and large-scale benchmarks**.
> - Across TinyImageNet-C, ImageNet-C, PACS, gradual shifts, and non-IID streaming, CRETTA consistently delivers higher accuracy, lower calibration error, and far better efficiency. Overall, the empirical evidence shows that CRETTA is robust, scalable, and reliable, providing meaningful improvements over strong baselines in realistic TTA scenarios.

---

> ### Author Response · Authors · 2025-11-21
>
> # Q2: Efficiency Advantages of CRETTA Over Sampling-Based Energy Methods
>
> Energy-based adaptation methods such as TEA incur substantial computational cost because they rely on iterative sampling procedures. This iterative scheme imposes a severe computational burden by requiring multiple rounds of forward and backward passes for every test batch resulting in excessive GFLOPs and memory usage. For instance, TEA reaches 4,335.82 GFLOPs on CIFAR10-C.
>
> In contrast, CRETTA is sampling-free, thereby completely eliminating the expensive gradient computation costs associated with iterative sampling. The only additional overhead is the marginal cost of forward passes required for source energy calculation. As shown in Table 10, this design results in only 527.40 GFLOPs, making CRETTA approximately 8times(8X) more efficient than TEA. Crucially, despite this significantly reduced computational cost, CRETTA demonstrates competitive performance, maintaining high classification accuracy and low Expected Calibration Error (ECE). Thus, our method attains comparable or better predictive performance at a fraction of the cost, making it substantially more practical for real-world deployment where efficiency and latency are critical.

---

> ### Author Response · Authors · 2025-11-21
>
> # Q3: Clarification on Table 9 Reference
>
> The reference to “Table 9” corresponds to the calibration results provided in the supplementary material. Due to the separation of the main manuscript and supplementary file during submission, the internal table linking was unintentionally disabled, causing the table to appear only in the supplementary document. We will correct this in the camera-ready version by explicitly referencing the supplementary table from the main text to avoid confusion.

---

### Official Review · Reviewer_ew8X · 2025-10-29

**Soundness:** 2
**Presentation:** 3
**Contribution:** 2
**Rating:** 4
**Confidence:** 4

**Summary:**

This paper introduces CRETTA, a sampling-free energy-based framework for Test-Time Adaptation (TTA). Unlike conventional TTA methods that rely on uncertain conditional predictions (e.g., entropy minimization) or costly energy-based sampling, CRETTA focuses on modeling only the residual energy, the discrepancy between the source and target distributions. By embedding this residual energy into a contrastive learning objective, CRETTA removes the need for normalization constant approximation or Markov Chain Monte Carlo (MCMC) sampling, achieving well-calibrated and efficient adaptation.

**Strengths:**

- The paper introduces a residual energy formulation that redefines TTA as learning only the distributional discrepancy between source and target domains. It is conceptually fresh and removes a reliance on normalization constant approximation.
- The paper is well-structured and readable.
- Experiments are extensive, covering CIFAR10/100-C, TinyImageNet-C, PACS, and ImageNet-C, with consistent improvements in both accuracy and calibration error (ECE).

**Weaknesses:**

- Although the paper argues that residual energy learning stabilizes adaptation, its claimed insensitivity to the source buffer suggests that the absolute source energy distribution plays a minor role. This raises the question of whether residual learning is fundamentally required, could similar stability be achieved simply by modulating target energies relative to arbitrary reference energies? (For example, AEA uses the low energy target samples as source buffer to reduce the source-target energy gap.)
- The ablation studies focus mainly on buffer composition and size; additional analysis on temperature sensitivity or else for residual learning could strengthen understanding of the method’s robustness.
- While the paper reduces the computational overhead of energy-based TTA by removing normalization constant estimation, the core idea of using EBMs for TTA follows prior works such as TEA and AEA. The contribution feels incremental, as sampling-free energy optimization has already been actively explored in other domains (e.g., sampling-free EBMs, RLHF, DPO). Moreover, performance gains over those baselines seem to be marginal.
- Evaluation is confined to standard online TTA; results under continual or episodic adaptation are missing, limiting the understanding of CRETTA’s robustness in dynamic environments.
- The ablation using CIFAR10-C with CIFAR100 as a replay buffer is not fully convincing, since the two datasets share similar data distributions and semantics. A more meaningful test would involve substituting with a cross-domain dataset (e.g., PACS or TinyImageNet) or reversing the setup (using CIFAR10 as the buffer for CIFAR100-C).

**Questions:**

See weakness section

---

> ### Author Response · Authors · 2025-11-21
>
> # W1&5: Role of residual learning and source energy
>
> > **The Importance of Residual Learning in Stable Adaptation**
>
> To assess whether residual learning is fundamentally necessary, we first examine how optimization changes when components associated with the residual learning framework are removed from our objective.
>
> In Appendix C, we analytically derive how the objective is altered when the residual component is excluded from our pair-wise contrastive formulation. Because the residual encodes the discrepancy between the source and target distributions, eliminating the residual framework makes the model no longer anchored to the source distribution. As a result, all terms involving the source model ($\phi$) vanish.
>
> In Table 15 (Appendix A.3.1), we evaluate the effect of removing the residual component by comparing classification accuracy and expected calibration error (ECE) between CRETTA (Default) and CRETTA (w.o. Source Model Term) at severity level 5 across three benchmark datasets. Removing the residual component consistently degrades both accuracy and calibration in most cases.
> Taken together, these findings validate the residual learning framework and incorporating the associated terms into our final objective is essential for stable and reliable adaptation.
>
> > **The Choice and Role of Source Energy Distribution in Contrastive Learning**
>
> Source energy (i.e., $E(x_s)$) is required not because of residual learning; but because it is a crucial component of the pairwise contrastive objective.
> Using an arbitrary reference energy as source energy would likely cause training failure. From a gradient perspective, Eq. (4) shows that during early adaptation a high source energy drives the gradient weight $w$ to collapse to zero, resulting in a trivial solution where learning cannot proceed—mirroring the well-known constraint in NCE regarding the choice of the noise distribution. To enable effective early adaptation, the buffer should therefore contain samples drawn from a distribution similar to the one learned by the pretrained source model, ensuring that these samples receive low energy and provide meaningful contrastive learning signals.
>
> **[Table A]** Performance Comparison of Source Buffer Contents on CIFAR100-C.
> | Buffer Type | Severity 5 | Severity 1–5 |
> |--------------------|------------|--------------|
> | **CRETTA (ours)**  | **64.52%** | **69.05%**   |
> | CIFAR-10 (train)   | 64.97%     | 69.37%       |
> | CIFAR-10 (val)     | 64.95%     | 69.37%       |
> | PACS (sketch)      | 63.74%     | 68.25%       |
>
> This behavior is confirmed empirically. As shown in Table A, CRETTA maintains strong performance on CIFAR100-C when the buffer contains CIFAR-10, which is not original source data, but distributionally similar. In contrast, performance drops when using a dataset with a very different distribution, such as PACS.
>
> Thus, Table 6 should not be interpreted as suggesting that the absolute source-energy distribution has a minor role. Instead, it shows that CRETTA remains robust and consistently effective as long as the buffer contains data that are distributionally similar to the source distribution-even without access to the original source dataset. In practice, while the true source data may be inaccessible due to privacy, obtaining similar samples is usually far more feasible.
>
> In summary, CRETTA’s superior performance does not stem merely from lowering the target energy. Rather, it arises from the integration between the residual formulation, the pair-wise contrastive objective, and their integration, which together provide a stable and powerful adaptation mechanism.

---

> ### Author Response · Authors · 2025-11-21
>
> # W2: Additional Analysis on temperature sensitivity and residual learning
>
> We provide additional analysis, experiments, and mathematical discussions in our Appendix. Specifically:
>
>
> > **Analysis on Temperature Sensitivity and Residual Learning**
>
> **Temperature sensitivity analysis (Figures 8 and 9 in Appendix)** demonstrates that our residual formulation remains stable across a wide range of temperatures, as it relies on relative residual energy differences rather than absolute energy values.
>
> **Loss Ablations (Table 15) and Gradient Ablations (Table 16)** show that the residual objective is the primary factor driving stable adaptation, consistently providing robust performance even when alternative loss components are removed or modified.
> > **Additional Experiments**
>
> **Non-i.i.d scenarios (Table 14)** show that our method remains effective under dynamic and non-stationary test-time environments.
>
> **Stable Adaptation over batches (Figure10,11,12)** further illustrates that maximizing the marginal likelihood of target samples maintains stable confidence compared to relying on uncertain predicted probabilities $p(y|x)$ in the test-time learning objective
>
> **Ablations on Buffer (Table17) and Pair Size (Table 18)** confirm that a contrastive learning framework is essential for stable and effective adaptation.
>
> Please let us know if you need any clarifications or further explanations regarding the experiments or discussions!

---

> ### Author Response · Authors · 2025-11-21
>
> # W3: Comparison with existing methods
>
> > **Contributions of CRETTA**
>
> By modeling the TTA problem through a residual energy formulation, CRETTA removes the normalization constant from the optimization process. This design choice is not only mathematically novel but also practically impactful: it addresses two longstanding scalability challenges in traditional EBM-based methods—
> (1) unresolvable approximation errors, and
> (2) prohibitively high computational cost.
>
> Crucially, while alleviating these issues, CRETTA remains well-calibrated across diverse distribution shifts, demonstrating both stability and robustness. Although the individual components of our method are familiar to the ML community, the novelty lies in their unique combination and tailored application to TTA—specifically designed to overcome the calibration limitations of existing approaches in real-world, performance-critical settings as acknowledged by reviewers fAWb and GnBf.
>
> > **Comparison to TEA**
>
> As shown in Figure 1, TEA collapses on large-scale datasets such as ImageNet-C and PACS. In the ImageNet-C experiment, TEA accumulates approximation error throughout adaptation, leading to progressively lower accuracy and higher ECE, ultimately reaching 31.60% accuracy / 8.39% ECE. In contrast, CRETTA attains 37.05% accuracy and 4.43% ECE, nearly halving the calibration error. This performance gap consistently appears on other large-scale datasets such as PACS as well (see Table 2 and Table 3).
>
> > **Comparison to AEA**
>
> AEA is designed for fast adaptation, accelerating updates by reducing the energy gap between target and source-like samples, which introduces an auxiliary loss on top of an entropy-minimization objective. However, its dependence on uncertain pseudo-labels means it inherits the core limitations of entropy minimization. As shown in Table 1, AEA’s calibration error is comparable to–or even worse than–TENT (e.g., on CIFAR100-C and TinyImageNet-C). Thus, despite its speed, AEA does not address the central challenge of uncertainty-aware adaptation in contrast to CRETTA, which provides well-calibrated adaptation through optimization over the marginal likelihood p(x).
> > **Comparison to Methods from Other Domains**
>
> SFT approaches such as DPO and RLHF originating from the language modeling domain, fundamentally rely on supervised signals: they require preference-labeled or ground-truth response pairs $(y_w, y_l)$ for a given context $x$. In contrast, CRETTA performs label-free adaptation to the target distribution in a TTA setting, aligning more closely with unsupervised learning.

---

> ### Author Response · Authors · 2025-11-21
>
> # W4: CRETTA in Dynamic Scenarios
>
> > **Dynamic Scenario1: Gradual Shifts**
>
> In Table 4 of the main paper, we show that CRETTA remains robust even when the target distribution progressively diverges from the source, outperforming TEA under gradual shifts. Moreover, while TEA suffers from degraded classification accuracy when the distribution eventually returns to the source domain—falling below its pre-adaptation performance—CRETTA continues to improve, increasing from 73.97% to 75.70%. This demonstrates that CRETTA’s residual interpretation implicitly provides a mechanism for catastrophic forgetting prevention, highlighting its potential for real-world deployment where distributions may evolve and later revert.
>
> > **Dynamic Scenario2: Non-i.i.d Data Streams**
>
> CRETTA’s robustness is further confirmed in our additional dynamic scenario experiments. In Appendix Table 14, we evaluate methods on CIFAR100-C (severity 5) under a realistic non-i.i.d. data stream. Even under severe class imbalance, CRETTA consistently outperforms all baselines and decisively surpasses the second-best method, TENT. These results illustrate CRETTA’s reliability in handling irregular, dynamically shifting data distributions.
>
> > **Dynamic Scenario3: Episodic Adaptation**
>
> We additionally assess how CRETTA and TENT perform in an episodic adaptation setting, where the model is reset at every adaptation step (evaluated on CIFAR10-C). CRETTA achieves higher accuracy under both the most challenging severities and the averaged severity metric, showing a notable gap in ECE. The results demonstrate CRETTA’s superior calibration even under repeated resets.
>
> **[Table F]** Performance Comparison of Episodic Adaptation Setting on CIFAR10-C.
> | | SEV5 ACC | SEV5 ECE | SEV1-5 ACC | SEV1-5 ECE |
> | :--- | :--- | :--- | :--- | :--- |
> | **TENT** | 85.60% | 6.40% | 89.23% | 4.41% |
> | **OURS** | **85.70%** | **5.39%** | **89.30%** | **3.55%** |
>
> Across gradually shifting distributions, non-i.i.d. data streams, and episodic adaptation scenarios, CRETTA consistently delivers strong performance. These results collectively validate CRETTA’s robustness in dynamic environments

---

> > ### Comment · Reviewer_ew8X · 2025-11-26
> >
> > I appreciate the authors for providing the detailed response.
> >
> > - Regarding W1, I cannot fully agree with the argument in the response. The authors state that “Eq. (4) shows that during early adaptation a high source energy drives the gradient weight to collapse to zero, ...”. However, this situation appears to be rather unusual, as the source energy does not necessarily need to be larger than that of the target samples. In such a regime, directly minimizing (or regularizing) the target energies would seem more straightforward and efficient than relying on source buffers. Moreover, Table A shows that even when using a totally different distributional dataset such as PACS, the performance degradation is only 0.79 (Sev. 5) and 0.8 (Sev. 1-5), suggesting that the source buffer is not a critical component for achieving residual learning. Therefore, my fundamental question remains: why and how is the proposed source buffer (or the pair-wise contrastive learning) exclusively necessary?
> > (c.f., I could not find the source for 'EpoTTA' in the first paragraph. Did you mean EpoTTA?)
> >
> > - Regarding the novelty of CRETTA, I agree that CRETTA’s empirical improvements in calibration are convincing and relevant to the TTA literature. However, I still consider that the methodological contributions appear incremental when compared to the existing works mentioned by the authors.
> >
> > Based on the reasons above, I will make my final decision and continue to monitor the other discussions (including comments from the other reviewers). Once again, thank you for the detailed response.

---

> ### Author Response · Authors · 2025-11-28
>
> # Why and How is contrastive learning exclusively necessary? (1/2)
> ---
> We appreciate your constructive feedback and the insightful points you raised, including those related to other weaknesses (e.g., hyperparameter ablation, dynamic scenarios). We hope that our earlier clarifications were helpful in addressing those concerns.
>
> After revisiting your comments, we realized that our previous response did not sufficiently resolve your primary question in W1, namely the necessity of contrastive learning in our method. Therefore, we would like to provide an extended explanation.
>
> - **(1) Why – Contrastive learning is necessary for well-calibrated and stable adaptation.**
>   We demonstrate this by **removing the contrastive term**, showing that direct minimization of target energy leads to unstable energy collapse and degraded calibration across benchmarks.
>
> - **(2) How – Contrastive learning provides a more informative gradient signal.**
>   We validate this by **replacing source-buffer samples with target samples**, showing that target-only regularization produces weak gradients and yields only marginal adaptation.
>
> ---
>
> ### **(1) Why – Contrastive learning is necessary for well-calibrated and stable adaptation**
>
> > First, to empirically validate whether contrastive learning is indeed essential for stable adaptation, we first conducted an additional experiment in which we removed the contrastive term and measured the resulting calibration error across the three benchmark datasets.
> >
> > **[Table B]** Ablations on the contrastive component across benchmark datasets. (ECE)
> | | CIFAR10-C Sev5 | CIFAR10-C Sev1-5 | CIFAR100-C Sev5 | CIFAR100-C Sev1-5 | TinyImageNet-C Sev5 | TinyImageNet-C Sev1-5 |
> | :--- | :--- | :--- | :--- | :--- | :--- | :--- |
> | **W Contrastive Terms (OURS)** | **4.15%** | **2.88%** | **7.99%** | **6.82%** | **13.52%** | **11.85%** |
> | **W.O Contrastive Terms** | 5.57% | 4.08% | 11.61% | 9.65% | 16.21% | 14.12% |
> >
> > As shown in Table B, removing the contrastive terms (i.e., $E(x_s)$) and directly minimizing the target energy led to a consistent degradation in calibration across all benchmarks. The effect was particularly pronounced on CIFAR100-C, where the calibration error deteriorated to 11.61% (+4%p), which is notably worse than TENT (8.93%) , an entropy minimization based method that is prone to overconfidence. These results clearly demonstrate that the stability of CRETTA’s adaptation does not arise simply from reducing target energies, as suggested in your comment, but instead stems from the contrastive learning mechanism.
> >
> > To further analyze how the contrastive term contributes to stable adaptation, we also examined the behavior of energy levels and calibration error on CIFAR100-C (Severity 5) during adaptation, comparing CRETTA against the variant where the contrastive terms are ablated.
>
> > **[Table C]** Target energy and ECE of CRETTA and without contrastive terms during adaptation on CIFAR100-C (Sev. 5).
> |  | Target Energy | | ECE | |
> |----------------|-----------------------------------|----------------|-----------------------------------|----------------|
> | **Batch Idx** | **W Contrastive Terms (OURS)**| **W.O Contrastive Terms** | **W Contrastive Terms (OURS)** | **W.O Contrastive Terms** |
> | 0 (First Batch) | -9.9781 | -9.9781 | 11.11% | 11.56% |
> | 9 | -10.1208 | -10.9536 | 10.48% | 11.64% |
> | 19 | -10.2124 | -11.5247 | 9.48% | 12.97% |
> | 29 | -10.1896 | -11.7863 | 9.85% | 12.35% |
> | 39 | -10.1856 | -12.0177 | 8.88% | 13.35% |
> | 49 (Last Batch) | -10.1904 | -12.2531 | 9.15% | 12.92% |
> | **$\Delta$ (Last - First)** | **-0.21** | **-2.28** | **-1.96%** | **1.35%** |
> >
> > As shown in Table C, we observe that when the contrastive terms are removed and the model directly minimizes the target energy, the energy level drops rapidly during the early stages of adaptation. While this may facilitate fast initial adaptation, it poses a critical risk to stability since aggressively lowering target energies early on sharpens the energy landscape around target samples, which can lead to overfitting. Empirically, we indeed find that removing the contrastive terms results in an overall increase in calibration error.
> >
> > In contrast, **CRETTA reduces the energy level progressively within the contrastive learning framework, enabling a more stable adaptation trajectory. This gradual reduction improves calibration error over time, demonstrating that the contrastive mechanism plays a key role in stabilizing the adaptation dynamics.**

---

> ### Author Response · Authors · 2025-11-28
>
> # Why and How is contrastive learning exclusively necessary? (2/2)
> ---
> ### **(2) How – Contrastive learning provides a more informative gradient signal**
>
> > Furthermore, the contrastive learning mechanism proposed in CRETTA that utilizes a small buffer of source (or distributionally similar) samples provides a more meaningful gradient signal than using target samples for regularization. In our previous comment, we illustrated this using an extreme scenario where the source energy, driven by high-energy target samples, becomes sufficiently large that the gradient effectively vanishes. Here, we provide a more realistic explanation focusing on the relative magnitudes of the energy level differences.
>
> > Target samples within the same batch are drawn from the same underlying distribution, and so thus it is unlikely for their energy levels to differ significantly. In contrast, source (or distributionally similar) samples originate from distribution that the pretrained model has already learned, making them more likely to exhibit consistently lower energy values than newly encountered target samples. **This creates a meaningful energy gap from the target sample energies, which in turn provides a strong gradient signal during adaptation. Consequently, CRETTA’s contrastive learning framework yields notable gains in classification performance while achieving better calibration.**
>
> > CRETTA provides a substantially more informative learning signal than simply replacing source samples with low-energy target samples. To validate this, we constructed an experimental setup where the 50% of samples with the lowest model-computed energy values in each target batch served as source-buffer data to compute the source energy $E(x_s)$. We then compared this setup with CRETTA’s adaptation process and performance.
>
>
> > **[Table D]** Gradient coefficient ($w$) of CRETTA and Target as Source Buffer Data during Adaptation on CIFAR100-C (Sev. 5).
> | Batch Idx | CRETTA | W TRG as SRC |
> |-----------|-------|-----------|
> | 0 | 0.490 | 0.304 |
> | 9 | 0.535 | 0.406 |
> | 19 | 0.580 | 0.456 |
> | 29 | 0.580 | 0.465 |
> | 39 | 0.627 | 0.518 |
> | 49 | 0.608 | 0.522 |
> | **AVG** | **0.572** | **0.454** |
>
>
> > **[Table E]** Performance Comparison of Target as Source Buffer Data on CIFAR100-C (Sev. 5).
> | Method | Sev5 Acc | Sev5 ECE |
> |----------------|----------------------|----------|
> | BN Adapt | 60.74% | 8.32% |
> | W TRG as SRC | 61.39% (+0.65%p) | 8.19% |
> | **CRETTA** | **64.52% (+3.78%p)** | **7.99%** |
>
>
> > More concretely, we first compared the magnitude of the gradient coefficient $w$ throughout the adaptation process. As shown in Table D, **replacing source samples with low energy target samples** results in consistently smaller gradient coefficients than CRETTA across the entire adaptation trajectory. This indicates that the model receives weaker learning signals and therefore **fails to sufficiently adapt to the target distribution**. Consequently, as shown in Table E, the accuracy improvement is only marginal amounting to just +0.65 percentage points compared to BN adapt, which performs adaptation solely through normalization without any learning. In contrast, **CRETTA maintains a relatively meaningful gradient coefficient while gradually increasing the learning signal from the early, high-uncertainty stages of adaptation toward later stages.** This leads to both improved classification performance and better calibration, ultimately achieving effective and stable adaptation.
>
> ---
>
> ### **Conclusion**
> > CRETTA’s contrastive learning is an essential component for achieving well-calibrated and stable test-time adaptation. By progressively lowering target energy and thereby reducing calibration error, it enables a stable adaptation process. Moreover, CRETTA’s contrastive learning methodology, which leverages buffer data, is distinctly more effective than approaches that apply only target-sample-based regularization. Notably, CRETTA’s contrastive framework is also robust to variations in buffer-data content and quality, making it highly practical for real-world deployment. For the results in Table A, if we focus not on the slight performance drop but on CRETTA’s consistent superiority over other baselines (e.g., TEA, EATA) even under the PACS-as-source setting, its robustness becomes even more evident. Taken together, CRETTA’s contrastive learning is thus a necessary and effective ingredient for achieving well-calibrated and stable adaptation.
>
> #### (We also apologize for the typographical error, “EpoTTA” should be “CRETTA”.)

---

> ### Author Response · Authors · 2025-11-28
>
> # Novelty of CRETTA
> ---
>
> We first thank the reviewer for acknowledging the strengths of CRETTA, especially regarding calibration which is indeed the central motivation behind our work. Additionally, we would like to clarify the methodological contributions of our approach.
>
> To the best of our knowledge, we are the **first to introduce a residual formulation for TTA** in the absence of ground-truth labels, whereby the adaptation task is redefined so that the model learns only the residual components that the pretrained model has not yet captured. Furthermore, this perspective also aligns with standard TTA practice that for stable adaptation, the backbone is typically frozen except for the BN layers and only a small subset of the model is updated. This small, selectively updated subset corresponds to the residual component adjusted at test time. Our formulation can thus be seen as extending this architectural design choice into an explicit learning-objective framework.
>
> Although the energy-based model (EBM) used in our framework may appear familiar, the key novelty of CRETTA does not lie in simply optimizing $p(x)$ utilizing an EBM. Instead, **our main contribution is that, within the residual learning framework, we integrate a contrastive learning objective that enables a simple mathematical trick to eliminate the normalization constant**, and devise a **computationally efficient yet reliable** form of test-time adaptation, one that existing adaptation strategies have not been able to achieve.
>
> Moreover, as you acknowledged, the effectiveness of our methodology is validated through extensive experiments: across datasets of varying scales (from CIFAR to ImageNet and PACS), diverse test-time settings (standard, gradual shift, non-iid and episodic), and comprehensive ablations (contrastive component–buffer, residual component–source model, temperature, etc.).
>
> We hope that these aspects of our work will be recognized, and sincerely appreciate you taking the time to engage in meaningful discussion to improve our work!

---

### Official Review · Reviewer_GnBf · 2025-11-01

**Soundness:** 3
**Presentation:** 2
**Contribution:** 3
**Rating:** 6
**Confidence:** 4

**Summary:**

This paper presents CRETTA, a  residual energy–based test-time adaptation framework designed to achieve efficient and well-calibrated adaptation under distribution shifts. Unlike entropy minimization–based methods that depend on unreliable pseudo-labels or energy-based approaches that require costly sampling, CRETTA introduces a residual energy function to model the discrepancy between source and target distributions. By embedding this residual function within a contrastive learning objective, the method removes the need for normalization constant approximation and significantly reduces computational overhead. Experiments across multiple benchmarks, including CIFAR10/100-C, TinyImageNet-C, PACS, and ImageNet-C, showing consistent improvements in both accuracy and calibration error, with strong robustness to overfitting and catastrophic forgetting.

**Strengths:**

1. The paper introduces a residual energy perspective* on test-time adaptation, which elegantly models distribution shifts as residual corrections to a pretrained energy landscape. This idea is both conceptually appealing and technically original, offering a clear advance over existing MLE- or entropy-based methods.


2. By eliminating sampling and normalization constant estimation, CRETTA achieves major computational savings (over 6× reduction in GFLOPs compared to TEA) without sacrificing performance, making it practical for real-time or resource-constrained deployment.


3. The experiments are thorough, covering multiple benchmarks and including ablation studies, buffer analysis, and gradual shift scenarios. The results convincingly demonstrate CRETTA’s robustness, calibration quality, and insensitivity to buffer composition.

**Weaknesses:**

1. The proposed framework adapts to the marginal distribution $p(x)$ via residual energy modeling, yet classification fundamentally depends on the conditional distribution $p(y|x)$. The paper does not clearly explain how aligning $p(x)$ leads to improved conditional decision boundaries or classification accuracy. Without a theoretical bridge (e.g., via Bayes decomposition or information-theoretic reasoning), the causal link between marginal alignment and better predictive performance remains speculative.

2. Dependence on source data: CRETTA relies on a small source buffer to perform contrastive adaptation. While the buffer can be as small as 1% of the source dataset and even substituted with similar-domain data, this still departs from the strict source-free TTA setting. In privacy-sensitive or memory-limited scenarios, this requirement might constrain the method’s deployment.

3. The contribution of the contrastive component is central to the method, yet there is no targeted ablation isolating its effect from the residual modeling itself. Including such analysis would help clarify whether performance gains stem mainly from contrastive optimization or other architectural factors.

**Questions:**

The proposed framework adapts to the marginal distribution $p(x)$ via residual energy modeling, yet classification fundamentally depends on the conditional distribution $p(y|x)$. Could the authors clarify how aligning $p(x)$ contributes to improved conditional decision boundaries and classification accuracy? Is there any theoretical justification (e.g., based on Bayes decomposition or information-theoretic reasoning) for this linkage?

---

> ### Author Response · Authors · 2025-11-21
>
> # W1&Q1 : Illustrations on the generalization performance of CRETTA
>
> We thank the reviewer for raising this important conceptual question. To clarify **how marginal alignment of $p(x)$ via residual energy modeling can lead to improvements in the conditional distribution $p(y \mid x)$**, we provide both a *(1) gradient-based interpretation* and an *(2) information-theoretic justification* to illuminate this linkage.
>
> ---
>
> We assume a classifier $f_\theta(x) = g(h_{\theta}(x))$ with feature extractor $h_{\theta}$ and a frozen linear classifier $g(z) = Wz + b, W = [w_1, \dots, w_C]^\top \in \mathbb{R}^{C \times d}$. The logits then can be expressed as $a_k(x) = w_k^\top h_\theta(x) + b_k$. Since the target domain contains no labels, we optimize the input density through the marginal energy, $E_\theta(x) = -\log \sum_{k=1}^C \exp(a_k(x))$ which is the standard unnormalized negative log-density in energy-based models. The gradient of this energy with respect to the logit $a_k(x)$ is given by $\frac{\partial E_\theta(x)}{\partial a_k(x)} = - \frac{\exp(a_k(x))}{\sum_{k'=1}^C \exp(a_{k'}(x))}=- p_\theta(y=k \mid x)$. Using $\frac{\partial a_k}{\partial z_t} = w_k$, the gradient of the energy with respect to the feature representation becomes $ \frac{\partial E_\theta(x)}{\partial z} = \sum_{k=1}^C \frac{\partial E_\theta(x)}{\partial a_k}\frac{\partial a_k}{\partial z} =- \sum_{k=1}^C p_\theta(y=k \mid x) w_k=- \mathbb{E}_{p _\theta(y\mid x)}[w_y]$.
>
> This expression shows that the gradient descent updates shift the feature $z$ toward the expected classifier weight vector. Although the classifier remains frozen, **this shift alters the logits and thus reshapes the conditional distribution, enabling the model to align its predictions to the target domain solely through feature-level adjustments**. Complementing this, from a feature perspective, energy minimization adapts the feature extractor to better capture the target-domain distribution producing stronger representations that improve classification accuracy under distribution shift even without labels.
>
> ---
>
> This can be also illustrated through mutual information. Let $I( ; )$ denote mutual information, and let $Z_{\text{pre}}$ and $Z_{\text{ada}}$ denote features extracted from the pretrained and the adapted model of a target sample respectively.
> The goal is to achieve $I(Y; Z_{\text{pre}}) < I(Y; Z_{\text{ada}})$.
> Since $I(Y; X_t)$ is fixed, comparing $I(Y; X_t \mid Z_{\text{pre}})$ and $I(Y; X_t \mid Z_{\text{ada}})$ is sufficient. Because energy minimization enables the adapted model to encode more of the structure of $X_t$, it is reasonable to expect $I(Y; X_t \mid Z_{\text{ada}}) < I(Y; X_t \mid Z_{\text{pre}})$, which directly implies a larger mutual information between labels and adapted features. Consequently, **learning $p(x)$ enhances the informativeness of the feature space and ultimately improves classification performance on the target domain.**
>
>
> We sincerely thank the reviewer for prompting this clarification and hope the expanded intuition helps convey the intended scope and theoretical grounding of our method.

---

> ### Author Response · Authors · 2025-11-21
>
> # W3: Ablations on contrastive component
>
> We thank the reviewer for this insightful question regarding the necessity of contrastive learning in CRETTA’s formulation.
>
> To evaluate the role of the contrastive component, we first conduct an experiment using only a single source sample in the buffer to minimize the role of the contrastive component preserving the structure of pairwise contrastive learning. **The results shown in Table17 (Appendix) show a clear degradation in both accuracy and ECE, highlighting the importance of the contrastive mechanism.**
>
> ---
>
> To further verify the role of contrastive components, we analyze the scenario in which the contrastive component is entirely removed. As shown by the gradient expression in Eq.(4), removing these terms eliminates the reactive regularization that stabilizes adaptation, as discussed in Lines 236-255.
>
> **[Table A]** Ablations on the contrastive component across benchmark datasets. (ECE)
> | | CIFAR10-C Sev5 | CIFAR10-C Sev 1-5 | CIFAR100-C Sev5 | CIFAR100-C Sev 1-5 | TinyImageNet-C Sev5 | TinyImageNet-C Sev 1-5 |
> | :--- | :--- | :--- | :--- | :--- | :--- | :--- |
> | **W Contrastive Terms(OURS)** | **4.15%** | **2.88%** | **7.99%** | **6.82%** | **13.52%** | **11.85%** |
> | **W.O Contrastive Terms** | 5.57% | 4.08% | 11.61% | 9.65% | 16.21% | 14.12% |
>
> As shown in Table A, removing the contrastive terms (i.e., $E(x_s)$) led to a consistent degradation in calibration across all three benchmark datasets, confirming its critical role in stabilizing adaptation. The effect was particularly pronounced on CIFAR100-C, where the calibration error deteriorated to 11.61% (+4%p), which is notably worse than TENT (8.93%) , an entropy minimization based method that is prone to overconfidence. These results clearly demonstrate that **the contrastive learning mechanism is a crucial component for achieving stability in CRETTA’s adaptation.**
>
> We appreciate the reviewer’s question, as it allowed us to clarify and strengthen this connection through expanded empirical and analytical evidence.

---

> ### Author Response · Authors · 2025-11-21
>
> # W2: Effectiveness of CRETTA in Source-Free Scenarios
> We would like to clarify the definition of source-free. Source-free refers to not using the data that was used during pretraining—that is, not accessing the original source dataset used to train the source model.
> As shown in Table 6, CRETTA remains effective even without access to the original source data, provided it has access to distributionally similar samples. Although real-world deployments often restrict access to source data (e.g., due to privacy), acquiring samples from a similar distribution is typically much easier, making CRETTA practical and robust in source-free settings. Moreover, in modern deployment environments, memory storage is relatively inexpensive compared to the computational burden of repeated sampling or approximating normalization constants. Therefore, using a small, static buffer is often far more practical than relying on computationally expensive alternatives (e.g., TEA).

---

### Official Review · Reviewer_fAWb · 2025-11-01

**Soundness:** 3
**Presentation:** 3
**Contribution:** 2
**Rating:** 6
**Confidence:** 3

**Summary:**

This paper introduces a test-time adaptation framework based on residual energy, CRETTA to enable efficient and well-calibrated adaptation under distribution shifts. In contrast to entropy-minimization methods that rely on unreliable pseudo-labels or energy-based approaches that demand expensive sampling, CRETTA employs a residual energy function to capture the discrepancy between source and target distributions. By integrating this residual function into a contrastive learning objective, the framework eliminates the need for normalization constant estimation and substantially reduces computational cost.

**Strengths:**

1. CRETTA avoids both sampling and normalization constant estimation, leading to remarkable efficiency gains relative to other energy-based method.

2. The residual design is well-motivated, it allows the model to adapt using minimal, controlled adjustments to the existing parameters, ensuring stability and preserving previously learned knowledge.

3. CRETTA consistently improves both accuracy and ECE across diverse datasets and corruption severities. The method maintains stable calibration even on challenging settings such as TinyImageNet-C and PACS, demonstrating that the proposed residual-energy mechanism contributes to reliable uncertainty estimation rather than merely higher accuracy.

4. The paper is well written and easy to follow, with logical organization and smooth transitions between motivation, method, and experiment.

**Weaknesses:**

1. The paper leverages an energy-based formulation for test-time adaptation, yet it remains unclear why energy modeling should be theoretically effective in this context. Could the authors provide more intuition or formal justification for why minimizing or adapting an energy function leads to improved generalization under distribution shift?

2. While the experiments cover standard corruption and small-to-medium-scale datasets (CIFAR10/100-C, TinyImageNet-C, PACS), the paper does not evaluate CRETTA on larger and more diverse domain generalization datasets such as DomainNet. Validation on such benchmarks would better demonstrate the scalability and robustness of the proposed method under complex, real-world domain shifts.

**Questions:**

Could the authors clarify the fundamental difference between energy-based and entropy-based test-time adaptation methods? Specifically, how does optimizing an energy function over the marginal distribution differ in objective and behavior from minimizing the prediction entropy of y given x?

---

> ### Author Response · Authors · 2025-11-21
>
> # W1 & Q1 : Clarifying Why Energy Minimization Improves Generalization and How It Differs from Entropy Minimization
> We thank the reviewer for raising questions that help clarify the theoretical foundations of how CRETTA operates. Below, we address *(1) why energy minimization improves generalization capability* and *(2) how this approach differs, especially from an optimization perspective, from entropy minimization* which is known to suffer from overconfidence issues.
>
> ---
>
> ### **Why does minimizing the energy function improve generalization under distribution shift?**
>
> > To understand why minimizing the energy of target samples leads to improved generalization capability (i.e., higher classification accuracy) on the target distribution, it is essential to examine how optimizing the energy function reshapes the representations. We assume a classifier $f_\theta(x) = g(h_{\theta}(x))$ with feature extractor $h_{\theta}$ and a frozen linear classifier $g(z) = Wz + b, W = [w_1, \dots, w_C]^\top \in \mathbb{R}^{C \times d}$. The logits then can be expressed as $a_k(x) = w_k^\top h_\theta(x) + b_k$. Since the target sample contains no labels, we optimize the marginal energy of the an input, $E_\theta(x) = -\log \sum_{k=1}^C \exp(a_k(x))$ which is the standard unnormalized negative log-density in energy-based models. The gradient of this energy with respect to the logit $a_k(x)$ is given by $\frac{\partial E_\theta(x)}{\partial a_k(x)} = - \frac{\exp(a_k(x))}{\sum_{k'=1}^C \exp(a_{k'}(x))}=- p_\theta(y=k \mid x)$. Using $\frac{\partial a_k}{\partial z} = w_k$, the gradient of the energy with respect to the feature representation becomes $ \frac{\partial E_\theta(x)}{\partial z} = \sum_{k=1}^C \frac{\partial E_\theta(x)}{\partial a_k}\frac{\partial a_k}{\partial z} =- \sum_{k=1}^C p_\theta(y=k \mid x) w_k=- \mathbb{E}_{p _\theta(y\mid x)}[w_y]$. This expression shows that the gradient descent updates shift the feature $z$ toward the expected classifier weight vector. Although the classifier remains frozen, **this shift alters the logits and thus reshapes the conditional distribution, enabling the model to align its predictions to the target domain solely through feature-level adjustments**. Complementing this, from a feature perspective, energy minimization adapts the feature extractor to better capture the target-domain distribution producing stronger representations that improve classification accuracy under distribution shift even without labels.
>
> ---
>
> ### **How does energy minimization differ from entropy minimization from an optimization perspective?**
>
> > To understand how two objectives behave differently during optimization, it is essential to compare their gradients. Since we already have expressed the gradient of the energy objective, we now present the gradient expressions for the entropy objective. For unlabeled target data, the entropy of the model prediction is expressed as $\mathcal{L}_ {\mathrm{ent}}(x) =- \sum_{k=1}^C p_k \log p_k$, where $p_k = p_\theta(y=k \mid x).$ The corresponding gradient is $ \frac{\partial \mathcal{L}_ {\mathrm{ent}}}{\partial z} = - \sum_ {y=1}^C (\log p_y + 1) p_y \left( w_y - \sum_ {k=1}^C p_k w_k \right)$. Letting $\mathbb{E}[w] = \sum_k p_k w_k$ denote the expectation of classifier weights under the current predictive distribution, we obtain the final compact form: $\frac{\partial \mathcal{L}_ {\mathrm{ent}}}{\partial z} = - \sum_ {y=1}^C (\log p_y + 1) p_y \left( w_y - \mathbb{E}[w] \right)$. In the gradient, the term $(\log p_y + 1)p_y$ heavily weights classes for which $p_y$ is already large, and $\log p_y$ is less negative. Thus, the gradient of entropy moves the feature $z$ in the direction that increases confidence for the most likely classes, effectively reducing prediction entropy and behaving similarly to pseudo-labeling. **By directly modifying the conditional distribution, entropy minimization mainly pushes the model to become more confident, often excessively.** In contrast, **the gradient of the energy depends only on $ \mathbb{E}_{p _\theta(y\mid x)}[w_y]$, which is a smooth expectation over classifier weights**. It does not contain the entropy term's confidence-amplifying multiplier. This trend is also empirically confirmed in Appendix Figures 10–12 as well.
>
> ---
>
> ### **Conclusion**
> - **Energy minimization improves generalization** by altering the logits and reshaping the conditional distribution, **allowing the model to align its predictions to the target domain**.
> - While achieving strong classification performance, it is also more robust than entropy minimization because it **optimizes a smooth expectation over classifier weights, avoiding the entropy objective’s confidence-amplifying multiplier**.
> - Together, these properties make energy-based adaptation a more balanced and principled optimization approach, enhancing representation quality rather than merely increasing confidence, which explains its **superior robustness in test-time adaptation**.

---

> ### Author Response · Authors · 2025-11-21
>
> # W2 : Scalability and Robustness on Large-Scale Benchmarks
>
> We thank the reviewer for the thoughtful suggestion to include evaluations on larger and more diverse domain generalization benchmarks such as DomainNet. We agree that such experiments could further illustrate the broader applicability of our method.
>
> We would like to clarify, however, that our work primarily targets the test-time adaptation rather than general domain generalization as it focuses on practical, real-world distribution shifts (e.g., clean → clean + snow corruption) rather than discrete and heterogeneous domain changes (e.g., paintings → photos).
>
> **[Table A]** Performance Comparison of TEA and CRETTA (Ours) on ImageNet-C.
> | | Sev5 acc | Sev5 ece | Sev1-5 acc | Sev1-5 ece |
> | :--- | :--- | :--- | :--- | :--- |
> | **TEA** | 31.60% | 8.39% | 38.72% | 7.21% |
> | **CRETTA** | **37.05%** | **4.43%** | **43.54%** | **2.69%** |
>
> Accordingly, we evaluate CRETTA across a wide range of TTA benchmarks with varying scales and complexities: CIFAR-10/100-C, TinyImageNet-C, and ImageNet-C. These datasets span diverse resolutions and label spaces, culminating in ImageNet-C, whose scale and class diversity are comparable to (or even greater than) DomainNet. As shown in Table A, **CRETTA’s strong performance on ImageNet-C provides evidence of its scalability large-scale and semantically complex data.**
>
> To demonstrate the method’s potential beyond TTA, we also include results on PACS, a standard DG benchmark, showing that CRETTA can naturally extend to this settings even though DG is not our main objective. While additional DomainNet experiments could further highlight DG applicability, we believe they are not essential for evaluating CRETTA within the TTA context, which is the core contribution of our paper. For future work, we consider extending relevant DG experiments, including DomainNet in the camera-ready version.
>
> We once again appreciate the reviewer’s perspective and hope this clarification helps outline the scope and intent of our study.

---

### Author Response · Authors · 2025-12-03
**We appreciate the reviewers’ valuable input and believe the concerns have been addressed as follows:**

**(Reviewer fAWb)**

> **Distinction between energy and entropy-based optimization**
>
> We compare the optimization behavior by computing the gradients of the two objectives. We showed that entropy minimization, due to its confidence-amplifying multiplier, drives the model to further increase its confidence, whereas energy minimization updates the target feature more smoothly without such effects. We also support this theoretical explanation with experimental results [1]. (*revision*)

> **Scalability evaluation using a large-scale and semantically diverse dataset**
>
> We demonstrate the scalability of CRETTA by evaluating classification accuracy and calibration error on ImageNet-C [2], a large-scale and diverse benchmark.

---

**(Reviewer GnBf)**
> **Robustness of CRETTA in the source-free setting**
>
> We clarify the definition of the source-free scenario and explain that our method achieves superior performance even when access to source data is restricted.

---

**(Reviewer ew8X)**
> **Ablation of residual formulation**
>
> We demonstrate that the residual learning formulation is a crucial component for stable adaptation by showing that removing relevant components leads to a performance drop with ablation results [3].

> **Additional Analysis on temperature sensitivity and residual learning**
>
> We present additional experiments: temperature-sensitivity analysis [4], residual-formulation[3] and gradient ablations [5], results on non-i.i.d. scenarios [6], visualizations of model confidence and calibration error over adaptation [1], and ablations on buffer [7] and pair size [8].

> **Additional Experiments in Dynamic Scenarios**
>
>We present additional experiments: gradual shifts [9], non-i.i.d. [6] and episodic [10] scenario, which collectively validate CRETTA’s robustness in dynamic environments.

---

**(Reviewer SoyK)**
> **Clarification of larger performance gain of CRETTA in Table 4 than Table 1.**
>
>We clarify that the modest gains in Table 1 are expected, as TEA’s approximation errors do not accumulate on small datasets and static settings, masking its inherent instability. When dataset size increases or adaptation scenario becomes dynamic, these errors compound and degrade TEA’s performance, whereas CRETTA remains stable, yielding substantially larger improvements on Table 4 and large-scale benchmarks [11, 12, 18,  6,  9].

---

**(Common)**
> **Clarification of why and how contrastive learning is essential for stable adaptation**
>
> We conduct an additional ablation comparing performance with and without the contrastive components, showing that the contrastive mechanism is essential for stable adaptation [14]. From an energy-based perspective, we find that CRETTA progressively decreases target energy and thus effectively reduces ECE [15].
>
> We further show that replacing the source data used for the contrastive term with low-energy target samples produces much weaker gradient signals, leading to insufficient adaptation. In contrast, CRETTA preserves a valid gradient coefficient, increasing the learning signal from early high-uncertainty to later stages [16, 17]. (*revision*)

> **Justification for how energy minimization and marginal distribution adaptation improve generalizability**
>
> We provide a theoretical explanation that the gradient of the energy w.r.t. the target feature, and the resulting update direction drives energy minimization to adjust logits, thereby reshaping the conditional distribution of the target data and improving classification accuracy. In addition, we offer an information-theoretic justification via mutual information, demonstrating that marginal alignment of $p(x)$ via residual energy modeling can enhance the conditional distribution. (*revision*)

> **Clarifying main contributions of CRETTA in test-time adaptation**
>
> We highlight CRETTA’s contributions unattainable in prior energy-based methods by contrasting their motivations and optimization behaviors. AEA focuses on rapid early updates but fails to sustain reliable adaptation, resulting in higher calibration error [12]. TEA improves calibration but suffers from normalization approximation errors on large-scale or dynamic scenarios and incurs heavy computational overhead from sampling [13]. In contrast, CRETTA closes the gap between calibration-aware adaptation and practical scalability: its residual formulation combined with a contrastive learning framework enables stable adaptation with substantially lower computational cost.

---


> 1. Figures 10–12 (Appendix)
> 2. Table A (Reviewer fAWb)
> 3. Table 15 (Appendix)
> 4. Figures 8–9 (Appendix)
> 5. Table 16 (Appendix)
> 6. Table 14 (Appendix)
> 7. Table 17 (Appendix)
> 8. Table 18 (Appendix)
> 9. Table 4
> 10. Table F (Reviewer ew8X)
> 11. Table A (Reviewer SoyK)
> 12. Table 1
> 13. Figure 2
> 14. Table B (Reviewer ew8X)
> 15. Table C (Reviewer ew8X)
> 16. Table D (Reviewer ew8X)
> 17. Table E (Reviewer ew8X)
> 18. Figure 1

---

### Author Response · Authors · 2025-12-03

### **Dear Area Chairs,**




> We sincerely appreciate your time and thoughtful evaluation. We fully understand and respect ICLR’s policy adjustments in light of this incident and thank you again for your consideration under a heavy workload. Since the review session was terminated earlier than expected, we were unable to receive rebuttal responses from three of the four assigned reviewers, and did not have adequate opportunity to engage in further discussion with the remaining reviewer to fully address his concerns. Accordingly, based on our best understanding of the reviewers’ concerns and intentions, we have articulated and reflected their points in our updated responses to each reviewer and in revision. We kindly ask that the ACs take this context into account when making the final decision, and we are grateful for your understanding.


---




### **Overview of Our Paper**




> - **Problem Setting: Test-Time Adaptation with Reliability Requirements**
>
> In this work, we aim to advance Test-Time Adaptation (TTA) not only by improving predictive performance but also by effectively enhancing model calibration, which is a critical aspect of reliability in practical deployment scenarios.


> - **Motivation and Limitations of Prior Work**
>
> Calibration is particularly important in environments where decisions have real-world consequences and must be made under strict latency or resource constraints. While entropy-minimization TTA adapts well to distribution shifts, they largely overlook calibration. Energy-based approaches have recently been proposed as promising alternatives for better calibration, but their reliance on normalization introduces unresolvable approximation errors and significant computational overhead, limiting their applicability in efficiency-critical scenarios. Or, they often fail to achieve reliable adaptation because of coupling with entropy minimization to preserve early-stage accuracy.
>


> - **How CRETTA Solves These Challenges**
>
> To address these challenges, we propose **CRETTA**. We reformulate TTA within the **residual framework** and integrate a **contrastive learning** objective that enables a simple mathematical trick to eliminate the normalization constant, and devise a **computationally efficient yet reliable** form of test-time adaptation.




> By bridging the gap between calibration-aware adaptation and practical feasibility, our approach offers a scalable solution previously unattainable with conventional TTA frameworks.




---




### **Strengths Highlighted by the Reviewers**
> Across the reviews, our work was recognized as introducing a **technically original residual energy formulation** (Reviewer fAWb, GnBf, ew8X, SoyK) coupled with a **contrastive objective that is methodologically well-motivated** (Reviewer SoyK) and empirically validated through **consistent improvements in accuracy and calibration** (Reviewer fAWb, GnBf, ew8X) across diverse datasets and challenging corruption settings (Reviewer fAWb, GnBf, ew8X), while simultaneously delivering **remarkable computational efficiency** (Reviewer fAWb, GnBf).




---




Once again, we thank all reviewers for their constructive feedback, the revisions have substantially strengthened the theoretical framing, empirical evidence, and clarity of CRETTA. We also thank the Area Chair for their time and consideration, and we hope this summary provides helpful context for your recommendation.




### **Best regards, The Authors.**

---

### Meta-Review · Area_Chair_sTph · 2025-12-27

**Summary:**

This paper proposes CRETTA, a sampling-free residual energy and contrastive objective for test-time adaptation, aiming to improve calibration and efficiency. While the motivation and experimental coverage are appreciated, reviewers raised concerns that the approach is incremental relative to prior energy-based TTA methods, with modest gains in standard settings and unclear overall significance. In addition, the paper exhibits an inconsistency with prior results on the PACS benchmark. Specifically, TEA underperforms TENT on PACS, which contradicts the findings reported in the original TEA paper, where TEA significantly outperformed TENT. While the authors provide a brief explanation for the low performance of TEA, this discrepancy is not discussed, raising concerns about experimental consistency and evaluation settings.
Based on these considerations, I recommend Reject. We appreciate the authors’ efforts in the rebuttal and revision, and encourage them to further refine the paper in line with the reviewers’ feedback.

**Reviewer Concerns:**

The rebuttal and revision partially addressed concerns by adding clarifications, ablations on the residual and contrastive components, and additional experiments on larger-scale and dynamic scenarios. However, key concerns remain regarding the methodological novelty, whether the contrastive/buffer design is essential relative to simpler alternatives, and whether the empirical improvements are sufficient to justify acceptance.

**Reviewer Scores:**

This submission received initial scores of 2, 4, 6, and 6. Given the initial average and that several key concerns remain partially unresolved after the rebuttal, I recommend Reject.

---

### Decision · Program_Chairs · 2026-01-26

Reject